# When Robustness Doesn't Promote Robustness: Synthetic vs. Natural Distribution Shifts on ImageNet

## Abstract

We conduct a large experimental comparison of various robustness metrics for image classification. The main question of our study is to what extent current synthetic robustness interventions ($\ell_p$-adversarial examples, noise corruptions, etc.) promote robustness under natural distribution shifts occuring in real data. To this end, we evaluate 147 ImageNet models under 199 different evaluation settings. We find that no current robustness intervention improves robustness on natural distribution shifts beyond a baseline given by standard models without a robustness intervention. The only exception is the use of larger training datasets, which provides a small increase in robustness on one natural distribution shift. Our results indicate that robustness improvements on real data may require new methodology and more evaluations on test sets representing natural distribution shifts.

## 1 Introduction

Reliable classification under distribution shift is still out of reach for current machine learning (Torralba et al., 2011; Recht et al., 2019). As a result, the research community has proposed a wide range of evaluation protocols that go beyond a single, static test set. Common examples include adversarial examples (Szegedy et al., 2013; Biggio et al., 2013), noise perturbations (Geirhos et al., 2018; Hendrycks & Dietterich, 2019), and spatial transformations (Fawzi & Frossard, 2015; Engstrom et al., 2019). Encouragingly, the past few years have seen substantial progress in robustness to these distribution shifts, e.g., see (Madry et al., 2018; Zhang et al., 2019; Geirhos et al., 2019; Zhang, 2019; Engstrom et al., 2019; Yang et al., 2019) and many others. However, an implicit assumption underlying this research direction is that robustness to such *synthetic* distribution shifts will lead to models that also perform more reliably on *natural* distribution shifts.

We challenge that assumption. We conduct a large experimental study involving 147 ImageNet models evaluated under 199 different evaluation settings for a total of 29,253 test set evaluations. Our model testbed contains a wide range of standard models and most proposed robustness interventions (adversarial training, various forms of data augmentation, etc.). In order to measure robustness under natural distribution shift, we utilize two recently proposed variants of ImageNet (Deng et al., 2009; Russakovsky et al., 2015) and ImageNetVid (Berg et al., 2015). Both test sets consist entirely of unperturbed images drawn from the same sources as the original datasets, but also exhibit substantial accuracy drops for all current model architectures (Recht et al., 2019; Shankar et al., 2019).

Figure 1 shows the main result of our evaluation. The plots display model accuracies for the two natural distribution shifts. As noted in prior work, there is a substantial drop in accuracy when going from the original test set to the test set with distribution shift. A priori, one may hope that a more robust model would see a smaller drop than baseline approaches without a robustness intervention. But we find that in both cases, current robustness interventions offer little to no benefit over standard models. In particular, the accuracy on the original test set alone almost perfectly predicts the accuracy on the test sets with distribution shift. Importantly, the same relationship between original and "shifted" accuracy holds for both standard models and models with an explicit robustness intervention. This implies that the robustness interventions do not close the gap between original and "shifted" test accuracies any more than standard models with the same accuracy.

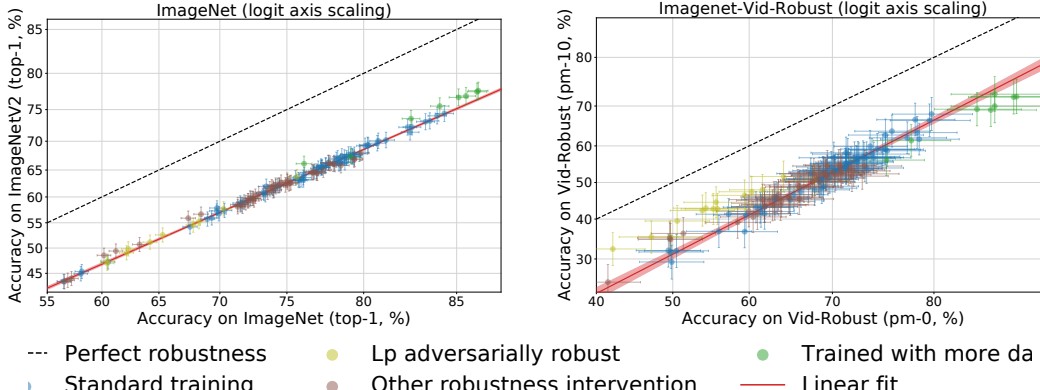

Figure 1: Model accuracy on the two natural distribution shifts, ImageNetV2 (left) and ImageNetVidRobust (right). Each data point corresponds to one model in our testbed and is shown with 99.5% Clopper-Pearson confidence intervals. The plots demonstrate that the standard test accuracy (x-axis) is an almost perfect predictor for the test accuracy under distribution shift (y-axis). This holds regardless of whether the model was trained with a robustness intervention. Current robustness interventions reduce the accuracy drops under these distribution shifts only by a small amount (on ImageNetVidRobust) or not at all (on ImageNetV2). The axes were adjusted using logit scaling and the linear fit was computed in the scaled space. The red shaded region is a 95% confidence region for the linear fit from 100,000 bootstrap samples.

There are only two significant deviations away from the otherwise universal relationship between original and shifted accuracy. The first is that $\ell_p$-adversarially robust models do offer increased accuracy against the distribution shift on ImageNetVidRobust. However, this increase is in a regime of overall low accuracy. As we will demonstrate in Section 5, interpolating between a standard model and a random classifier yields a comparable robustness increase in this accuracy regime. So the current $\ell_p$-robust models do not provide a truly new robustness trade-off.

The second class of outliers are models trained on substantially different data. In particular, the two most pronounced outliers on ImageNetV2 correspond to models trained on $10\times$ to $1{,}000\times$ more training examples. This shows that adding more training data does indeed increase robustness on ImageNetV2 more than any other currently proposed explicit robustness intervention. However, adding more data also yields only small gains: the accuracy drops of around 10% from ImageNet to ImageNetV2 shrink by $1 - 2\%$, even when adding $1{,}000\times$ more data.

Overall, our results show that current robustness gains on synthetic distribution shifts do not transfer to improved robustness on the natural distribution shifts presently available as test sets. This suggests that research on reliable machine learning may currently be focusing on interventions that do not promote robustness on natural distribution shifts. Achieving robustness on real data may instead require new methodology and more evaluations on natural distribution shifts.

To aid this development, we will release and maintain our robustness testbed as a platform for proposing and evaluating new models and datasets. In total, our testbed includes prediction data for 29,253,000 pairs of models and image inputs. We hope that a comprehensive repository for robustness evaluations will simplify the process of comparing proposed models and evaluation settings and enable further analysis of robustness questions.

In the next section, we describe our experimental setup in detail. Section 3 introduces our main measure of robustness. The following three sections then investigate our three main questions:

- Are synthetic robustness measures predictive of performance on natural distribution shifts?
- What robustness interventions are effective for natural distribution shifts?
- How does the amount of training data impact robustness?

## 2 EXPERIMENTAL SETUP

We first formally define our experimental setup. In the standard classification setting, a model $\hat{f}$ is first trained on i.i.d. samples from a fixed data distribution. We then test $\hat{f}$ on another set $S_1$ of i.i.d.

samples from the same distribution to compute the test accuracy $\mathrm{acc}_{S_1}(\hat{f}) = \sum_{(x,y) \in S_1} \mathbb{I}[\hat{f}(x) = y]$. We refer to any deviation from this setup as a *distribution shift*.

More specifically, our robustness evaluation involves distribution shifts defined as follows. Instead of using the original test set $S_1$, we may also evaluate the accuracy on a new test set $S_2$ that is collected via a similar but not identical procedure to the original test set $S_1$ (i.e., $S_2$ represents a different distribution). Moreover, we allow each data point (either from $S_1$ or $S_2$) to be pertubed before passing it to the classifier $\hat{f}$. We consider a general setup where the perturbation $\pi(x, y, \hat{f})$ may depend on the model $\hat{f}$ to be evaluated. For instance, we can generate $\ell_\infty$-adversarial examples by setting $\pi(x, y, \hat{f}) = \arg\min_{\|x-x'\|_\infty \leq \varepsilon} \mathbb{I}[\hat{f}(x') = y]$. Alternatively, we can simply perturb each data point with Gaussian noise, i.e., $\pi(x, y, \hat{f}) = x + \mathcal{N}(0, \sigma^2)$ independently of $y$ and $\hat{f}$.

Combining the choice of test set $S$ and perturbation $\pi$, the *test accuracy under distribution shift $S$, $\pi$* is then defined as

$$\mathrm{acc}_{S,\pi}(\hat{f}) = \sum_{(x,y) \in S} \mathbb{I}[\hat{f}(\pi(x, y, \hat{f})) = y] \ .$$

Depending on $S$ and $\pi$, this "shifted" test accuracy may be higher or lower than the original accuracy $\mathrm{acc}_{S_1}$. The focus of this paper is on distribution shifts where there is reasonable hope to classify the images correctly (e.g., because humans can do so), but standard models achieve only a substantially lower accuracy than on the original test set. Reducing these accuracy gaps between original and shifted accuracy is an important goal of robust machine learning.

## 2.1 TYPES OF DISTRIBUTION SHIFTS

At a high level, we distinguish between two main types of distribution shift. The crucial distinction between the two is whether the distribution shift involves a synthetic intervention at the pixel level. We use the term *natural* distribution shift for cases that rely only on unmodified images. In contrast, we refer to distribution shifts as *synthetic* if they involve modifications of existing test images. To be concrete, we now provide an overview of the various distribution shifts in our robustness evaluation.

### 2.1.1 NATURAL DISTRIBUTION SHIFTS

Our testbed includes two distribution shifts that involve only unmodified natural images.

**ImageNetV2.** The first example is the new ImageNetV2 test set recently collected by Recht et al. (2019). The goal of this test set was to closely reproduce the original ImageNet dataset creation process, e.g., by matching the data source (Flickr) and the data cleaning process (Mechanical Turk). Nevertheless, the authors found that state-of-the-art models still suffer a performance drop of at least 11% on the new test set. This makes the dataset an interesting example for studying robustness to natural distribution shifts. Testing on ImageNetV2 fits into our evaluation framework by letting $S$ be the new ImageNetV2 test set (denoted by $S_2$) and leaving $\pi$ to be the identity function.

**ImageNetVidRobust.** In contrast to precisely defined, synthetic notions of robustness such as $\ell_p$-adversarial examples, it is currently an open problem to characterize the distribution shift arising in ImageNetV2. To narrow this gap between the real and synthetic robustness challenges, Shankar et al. (2019) have introduced a variant of the ImageNetVid dataset (Berg et al., 2015). The authors assembled consecutive video frames that are all highly similar to a human yet cause classifiers to make errors on some of the images. Inspired by $\ell_p$-adversarial examples, the authors propose the following $\mathrm{pm}_k$ (plus-minus $k$) metric to evaluate the accuracy of a classifier as

$$\mathrm{pm}_{k,S}(\hat{f}) = \sum_{(x,y) \in S} \min_{x' \in \mathcal{B}_k(x)} \mathbb{I}[\hat{f}(x') = y] \ ,$$

where $\mathcal{B}_k(x)$ contains the $2k$ frames around the anchor frame $x$ ($k$ frames in each temporal direction). This corresponds to a perturbation function $\pi(x, y, \hat{f}) = \arg\min_{x' \in \mathcal{B}_k(x)} \mathbb{I}[\hat{f}(x') = y]$. Under this metric, the authors find that even the best models see an accuracy drop of at least 13%.

### 2.1.2 Synthetic Distribution Shifts

The research community has developed a wide range of synthetic robustness notions for image classification over the past five years. In our study, we consider the following classes of synthetic distribution shifts.

**Adversarial examples.** One prominent example of synthetic distribution shifts are adversarial examples, which demonstrate that current image classifiers can be fooled by small image perturbations that are (almost) imperceptible to a human (Szegedy et al., 2013; Biggio & Roli, 2018). In our robustness evaluation, we focus on untargeted adversarial perturbations bounded in $\ell_\infty$- or $\ell_2$-norm.

**Image corruptions.** Since $\ell_p$-adversarial examples are unlikely to occur outside a truly worst-case setting, the research community has proposed various synthetic image corruptions as less adversarial distribution shifts. The goal of these corruptions is to test robustness to distribution shifts that are more realistic and hopefully predictive of performance on real data (we will investigate this assumption in Section 4). In our testbed, we include all corruptions from (Hendrycks & Dietterich, 2019) and additionally some corruptions from (Geirhos et al., 2018). These include common examples of image noise (Gaussian, shot noise, etc.), various blurs (Gaussian, motion), simulated weather conditions (fog, snow), and "digital" corruptions such as various JPEG compression levels. We refer the reader to Appendix A.6.3 for a full list of the 38 corruptions.

**Style transfer.** We also include a style transfer version of the ImageNet test set (Huang & Belongie, 2017). This distribution shift was proposed to evaluate whether classification models are relying more on shape or texture features (Geirhos et al., 2019).

### 2.2 Classification Models

Our model testbed includes 147 ImageNet models covering a variety of different architectures and training methodologies. Our goal was to include most relevant pretrained ImageNet models available online. In multiple cases we also contacted paper authors with various degrees of success. At a high level, the models can be divided into the following three categories (see Appendix A.5.1 for a list of all models).

**Standard models.** We refer to models trained on the ILSVRC 2012 training set without a specific robustness focus as *standard models*. This category includes 74 models, ranging from the seminal AlexNet with top-1 accuracy 56.5% through widely used architectures such as VGG, ResNet, and Inception to the state-of-the-art EfficientNet with top-1 accuracy 84.4% (Krizhevsky et al., 2012; Simonyan & Zisserman, 2014; He et al., 2016; Szegedy et al., 2015; Tan & Le, 2019).

**Robust models.** This category includes models with an explicit robustness intervention. We subdivide this class further into two types of robustness interventions:

- Models with increased adversarial robustness. We include both models trained with projected gradient descent (Xie et al., 2019; Shafahi et al., 2019) and models with evaluation via randomized smoothing (Cohen et al., 2019; Salman et al., 2019), which yields a total of 14 models.
- Further robustness interventions such as data augmentation schemes (Geirhos et al., 2019; Yun et al., 2019) and anti-aliasing (Zhang, 2019). This subset contains 51 models.

**Models trained on more data.** Finally, our testbed also contains three types of models that utilized substantially more training data than the aforementioned models. The ResNet152-ImageNet11k model was trained on 12.4 million images for 11,221 classes from the full ImageNet dataset (Wu, 2016). Like all other models, we only evaluate it over the 1,000 classes from ILSVRC 2012. Facebook recently released models trained on 1 billion images from Instagram; we refer to these models as Instagram models (Mahajan et al., 2018). We also include evaluations from the JFT-300M model trained on 300 million images at Google (Sun et al., 2017).

## 3 Measuring the Effect of Robustness Interventions

When comparing interventions to increase robustness, a crucial question is how to measure their effect in a meaningful way. One approach would be to simply use the absolute accuracy numbers $\text{acc}_{S,\pi}$ under the distribution shift specified by $S$ and $\pi$ (see Section 2 above). If the goal is to select the best performing model (e.g., for a concrete deployment), this is indeed a relevant criterion.

On the other hand, relying on absolute accuracy numbers becomes less relevant when comparing various robustness interventions with different unperturbed accuracies $\text{acc}_{S_1}$. As can be seen from Figure 1, the accuracy under distribution shift varies substantially with the standard accuracy: models with higher standard accuracy already achieve higher accuracy under distribution shift. So to isolate the effect of a training intervention on the robustness to distribution shift, we would like to measure how much a robustness intervention increases the shifted accuracy *beyond what is possible with standard models at the same standard accuracy*.[1]

To quantify this notion of robustness, we define a "baseline" function $\beta_{S,\pi}(x)$ to return the shifted accuracy that a standard model with test accuracy $x$ on the standard test set $S_1$ is expected to achieve under the distribution shift $S, \pi$. So if $\hat{f}$ is a model without robustness intervention, $\beta$ should satisfy $\beta_{S,\pi}(\text{acc}_{S_1}(\hat{f})) = \text{acc}_{S,\pi}(\hat{f})$.

A priori, it is unclear if such a function $\beta$ exists. The accuracy of a trained model $\hat{f}$ on the distribution shift $(S, \pi)$ could depend on a variety of properties besides the standard accuracy of $\hat{f}$ (model architecture, data augmentation scheme used during training, etc.). However, both natural distribution shifts we study in our paper have the intriguing property that the standard test accuracy is a nearly perfect predictor of the test accuracy under distribution shift for all standard models in our testbed. In particular, fitting a simple logistic model to the $(\text{acc}_{S_1}(\hat{f}), \text{acc}_{S,\pi}(\hat{f}))$ pairs yields a good linear fit in the logit domain (see Figure 1). This shows that we can obtain an accurate estimate $\hat{\beta}$ from data, which we will rely on in the remainder of the paper.

With the baseline robustness function $\beta$ in place, we can now define the *effective robustness* $\rho(\hat{f}, S, \pi)$ of a trained model $\hat{f}$ to distribution shift $(S, \pi)$ as

$$\rho(\hat{f}, S, \pi) \ = \ \text{acc}_{S,\pi}(\hat{f}) - \beta_{S,\pi}(\text{acc}_{S_1}(\hat{f})) \ ,$$

If a model $\hat{f}$ is truly more robust under distribution shift $S, \pi$ (in particular, more so than standard models without a robustness intervention), $\hat{f}$ should have effective robustness $\rho(\hat{f}, S, \pi)$ substantially larger than 0. Graphically, $\rho$ can be interpreted as the vertical deviation away from the linear fit in Figure 1. In the following, we will study to what extent various synthetic robustness measures are predictive of the effective robustness on natural distribution shifts, and which models achieve the largest effective robustness on natural distribution shifts.

## 4 DOES SYNTHETIC ROBUSTNESS PREDICT NATURAL ROBUSTNESS?

Given the difficulty of measuring a model's robustness to natural distribution shifts, an important question is whether there are simple synthetic proxies. We therefore study to what extent robustness to synthetic distribution shifts predicts robustness on natural distribution shift.

A simple approach would be to directly compare different models' robustness to synthetic distribution shift and their robustness to natural distribution shift. However, we run into an important confounding factor. As mentioned earlier, models that have higher accuracy have a smaller accuracy drop on the natural distribution shifts we consider. Moreover, we find that models with higher accuracy also see smaller accuracy drops under various synthetic distribution shifts. Therefore it would not be methodologically sound to directly compare synthetic robustness to natural robustness and draw any conclusions about transferability between the two. Simply the fact that a model is more accurate will simultaneously predict both that (i) the accuracy drop under natural distribution shift will be smaller, and that (ii) the model will be more robust to synthetic distribution shift.

To control for the confounder standard accuracy, we rely on the effective robustness notion introduced in Section 3, i.e., the robustness increase that cannot be attributed to an increase in standard accuracy. In Figure 2, we analyze the relationship between two synthetic robustness notions and effective robustness on ImageNetV2. The resulting plot shows that the two quantities are almost entirely uncorrelated: a model being more robust to synthetic perturbations does not imply that the model will have a smaller accuracy drop on ImageNetV2. In Appendix A.1, we repeat the above experiment for the ImageNetVidRobust distribution shift and reach similar conclusions.

---

[1]We note that this corresponds to the problem of estimating a treatment effect as studied in causal inference. We do not explicitly rely on knowledge about causal inference to make our exposition more accessible but refer the reader to textbooks on causal inference for background (Pearl et al., 2016; Hernan & Robins, 2019).

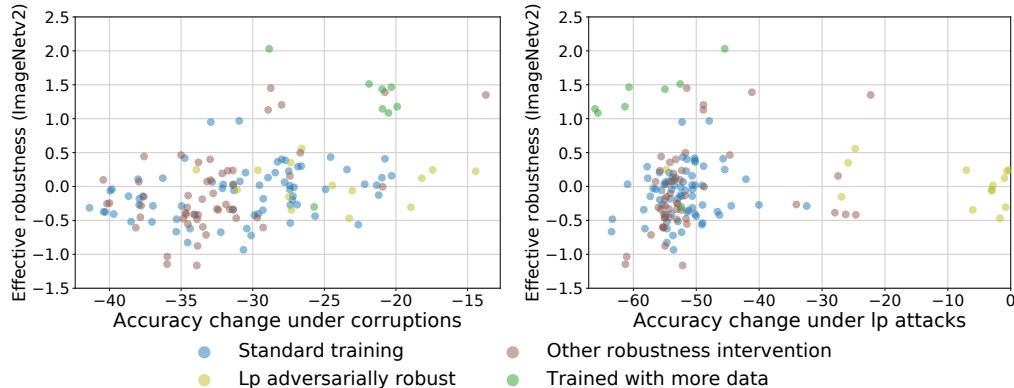

Figure 2: Robustness to synthetic distribution shift (x-axis) vs. effective robustness to the ImageNetV2 distribution shift (y-axis). The left plot shows synthetic robustness as measured by an average over image corruptions from (Hendrycks & Dietterich, 2019; Geirhos et al., 2018; 2019). The right plot shows synthetic robustness as measured by PGD attacks for $\ell_2$- and $\ell_\infty$-robustness and two perturbation sizes each. In both cases, synthetic robustness is not predictive of effective robustness under the natural distribution shift. Appendices A.1 and A.2 contains similar plots for individual synthetic robustness measures and the ImageNetVidRobust distribution shift.

## 5 WHAT HELPS WITH NATURAL DISTRIBUTION SHIFT?

As can be seen in Figure 1, most models did not achieve substantial effective robustness on ImageNetV2 and ImageNetVidRobust. Nevertheless, a few model types stood out with significant effective robustness on one of the two datasets (no model in our testbed that showed non-trivial gains for both natural distrbution shifts). We now discuss the main outliers in more detail.

**ResNet152-ImageNet11k and Instagram models.** The left plot in Figure 3 zooms to the two model families that displayed high effective robustness on ImageNetV2. All of these models are trained on additional data beyond the standard ILSVRC 2012 training set. The models exhibited an effective robustness of 2.0 and 1.4 (median robustness for the four Instagram models), which was substantially larger than any other model with similar accuracy. However, despite their promising performance on ImageNetV2, these models show little to no effective robustness for ImageNetVidRobust. Moreover, the effective robustness gain is larger for the ResNet152-ImageNet11k model trained on *less* data than the Instagram models. To further investigate the effect of data on model robustness, we conduct a series of controlled experiments in Section 6.

$\ell_p$**-robust models.** In the right plot of Figure 1 we see a cluster of $\ell_p$-adversarially robust classifiers that all exhibit substantial effective robustness for ImageNetVidRobust. However, the models are also in a low accuracy regime, in particular lower than the standard classifiers they were derived from (this trade-off is discussed in (Tsipras et al., 2019; Raghunathan et al., 2019) among others).

This trade-off poses a potential confounding factor: although the $\ell_p$-robust models show high effective robustness, it is not clear if the resulting accuracy trade-off is meaningful. To put the trade-off into context, we compare the models to a simple baseline. In particular, consider the family of classifiers given by interpolating between random guessing and the standard classifier the $\ell_p$-robust models are derived from. Since random guessing is not affected by the distribution shift, this interpolation can trade off standard accuracy for effective robustness. In the right plot of Figure 3, we illustrate this family of interpolated classifiers (the dotted line) and show that a standard ResNet152 with no robustness interventions can be interpolated with a random classifier to achieve the same trade-off as an $\ell_p$-robust model. So for the purpose of the natural distribution shift on ImageNetVidRobust, the $\ell_p$-robust models offer no benefit beyond interpolating with a random classifier.

## 6 HOW DOES THE AMOUNT OF TRAINING DATA IMPACT ROBUSTNESS?

In the previous section we have seen that the ResNet152-ImageNet11k and Instagram models achieve non-zero effective robustness on ImageNetV2. This is a plausible effect since larger training sets provide a more thorough sampling of real world images. However, the JFT-300M model does not display this effect despite being trained on $300\times$ more data than standard models.

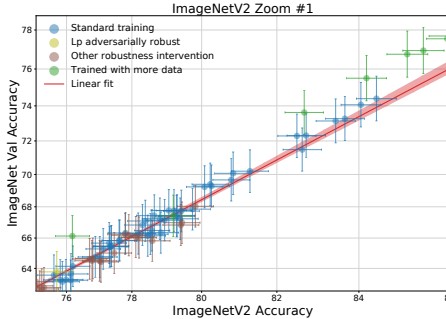 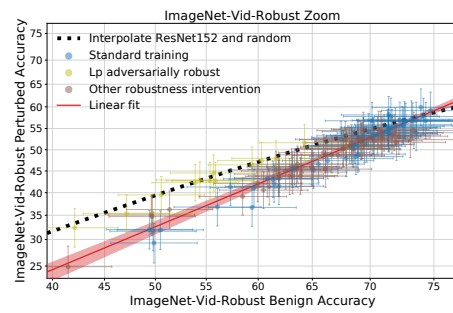

Figure 3: The left plot zooms into the range of the ResNet152-ImageNet11k and Instagram models on ImageNetV2 and shows that the models achieve significant effective robustness, i.e., they lie significantly above the linear fit. The right plot zooms into the range of the $\ell_p$-robust models on ImageNetVidRobust with high effective robustness. The dotted line shows the family of models achievable by interpolating between a standard ResNet-152 and a random classifier. It achieves the same robustness trade-off as the $\ell_p$-robust models.

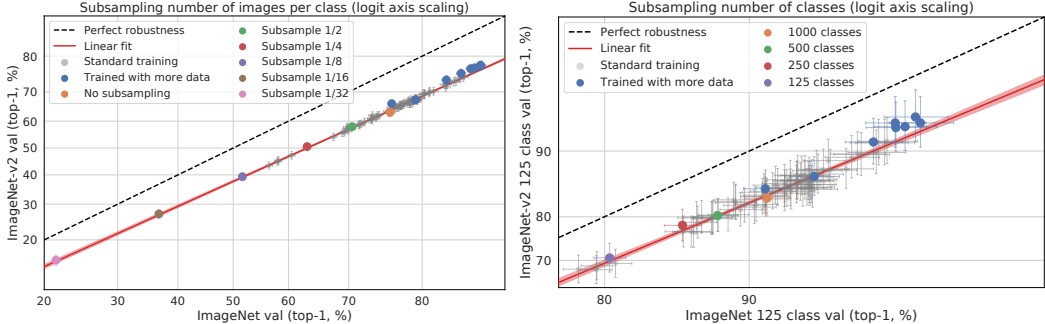

Figure 4: To investigate the impact of training data on robustness, we vary the ILSRVC 2012 data along two axes: the number of images per class (left), and the number of classes (right). Although models trained on more data (e.g., the Instagram and ResNet152-ImageNet11k models) provide improvements in effective robustness, we find that subsampling of the ILSVRC training has no impact on effective robustness.

A possible explanation could be that differences in label diversity or quality play a role in promoting robustness. For example, the Instagram models are trained on a dataset collected to overlap with the ILSVRC categories, and the ResNet152-ImageNet11k model is trained on a superset of the ILSVRC data. Meanwhile, the JFT model is trained on a long-tail, weakly-labeled dataset with 18,291 categories, which, to the best of our knowledge, were not collected to align with ILSVRC categories. To investigate the role of data in more detail, we now conduct two experiments.

**Varying the number of images per class.** We start by subsampling the number of images per class in the ILSVRC training set by factors of $\{2, 4, 8, 16, 32\}$ and show the impact on accuracy and robustness on ImageNetV2 in Figure 4. As expected, increasing the number of images per class consistently leads to higher accuracy. However, the impact on robustness is exactly as predicted by our fit on standard ImageNet models. This indicates that varying the the size of training sets in an i.i.d. manner affects accuracy but not effective robustness, at least for ImageNetV2.

**Varying the number of classes.** Next, we consider the *diversity* of the data by varying the set of classes. In particular, we create three successive subsets of the ILSVRC training set with 500, 250, and 125 classes (see Appendix A.6.5 for the list of classes). We then evaluate all models in our testbed on the 125 classes present in all subsets. We present results in Figure 4, which shows that varying the number of classes again affects accuracies but not effective robustness.

Our experiments suggest that neither growing the number of images per class nor the number of classes in an i.i.d. fashion are effective robustness interventions. Nevertheless, Section 5 shows that larger dataset can provide meaningful robustness improvements. This disparity may be due to limitations of emulating changes in the training set by subsampling ILSVRC. For one, our experiments consider i.i.d. subsets of the training images or classes and do not consider growing datasets in a

manner that may cover different image distributions, as in the case of Instagram data. Another possibility is that increases in dataset size may only improve robustness after the dataset size is large enough such that the accuracy on the original distribution is close to saturated. Our experiments only observe dataset sizes smaller than ILSVRC, which may fall below this inflection point.

## 7 RELATED WORK

There are several related papers that study the relationship between various synthetic robustness measure, e.g., (Geirhos et al., 2018; Engstrom et al., 2019; Hendrycks & Dietterich, 2019; Kang et al., 2019; Tramer & Boneh, 2019; Maini et al., 2019). To the best of our knowledge, our paper is the first to compare synthetic and *natural* distribution shifts. Since our findings show that synthetic robustness currently does not promote natural robustness, our paper offers a complementary view to the aforementioned comparisons focused on synthetic robustness.

Our work relies on two recent papers introducing datasets with natural distribution shifts (Recht et al., 2019; Shankar et al., 2019). Recht et al. (2019) already note that there is a clear linear relationship between original and new test set accuracy but do not evaluate any models with a robustness intervention or models trained on more data. Shankar et al. (2019) compare only a small set of robust models. Due to the limited size of their test set and model testbed, their results do not conclusively show that $\ell_p$-robust models offer effective robustness on their dataset. In contrast, our testbed of robustness interventions is substantially larger and aims to encompass all publicly available models. We also include models trained on more data and conduct experiments to investigate the effect of training data on robustness. While our experiments shows that $\ell_p$-robust models do offer effective robustness on ImageNetVidRobust, we also put this improvement in context by demonstrating that it is comparable to interpolating with a random classifier. Moreover, none of the two papers study to what extent synthetic robustness is predictive of robustness under natural distribution shifts.

## 8 CONCLUSION

One aspiration of robustness interventions is to improve performance on real distribution shifts. Since this is a challenging problem, synthetic notions of robustness offer a natural starting point. While it is expected that corresponding interventions will be less effective on real distribution shifts, the hope is that these interventions still convey at least some robustness on real data. Unfortunately, our experiments show that this is not the case for two recent examples of real distribution shifts. On both datasets, models with a robustness intervention perform no better than baselines without such interventions. The fact that progress on synthetic distribution shifts is essentially uncorrelated with progress on real distribution shifts raises the question whether further improvements for synthetic robustness will yield benefits on real data.

To deploy machine learning in safety-critical environments, it will be necessary to address this question and improve robustness under real distribution shifts. Our experiments suggests multiple directions for future work to achieve this goal.

One important direction is to assemble more examples of natural distribution shifts in order to test robustness on real data in more detail. While it is possible that other types of natural distribution shifts are more correlated with synthetic robustness, this hypothesis needs to be tested with carefully curated datasets.

Our experiments on ImageNetV2 suggest that adding training data can improve robustness. A similar phenomenon has also been demonstrated for adversarial robustness (Schmidt et al., 2018; Carmon et al., 2019; Uesato et al., 2019; Najafi et al., 2019; Zhai et al., 2019). However, it is currently unclear what type of data to add: the JFT-300M model trained on $300\times$ more data offers no robustness improvement. The ResNet152-ImageNet11k model was trained on roughly $10\times$ more data and offers a robustness gain comparable or larger than the Instagram models trained on $1,000\times$ more data. Understanding what additional training data is most effective could significantly improve robustness to real distribution shifts.

Finally, it is important to note that the synthetic robustness interventions do offer robustness on the distribution shifts they are designed for. This raises the hope that accurately characterizing real distribution shifts will allow us to leverage existing techniques for training more robust models under well-specified distribution shifts. The distribution shifts in our comparison and our extensive model testbed offer a starting point for such an investigation into improved training methodology.

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

# A   APPENDIX

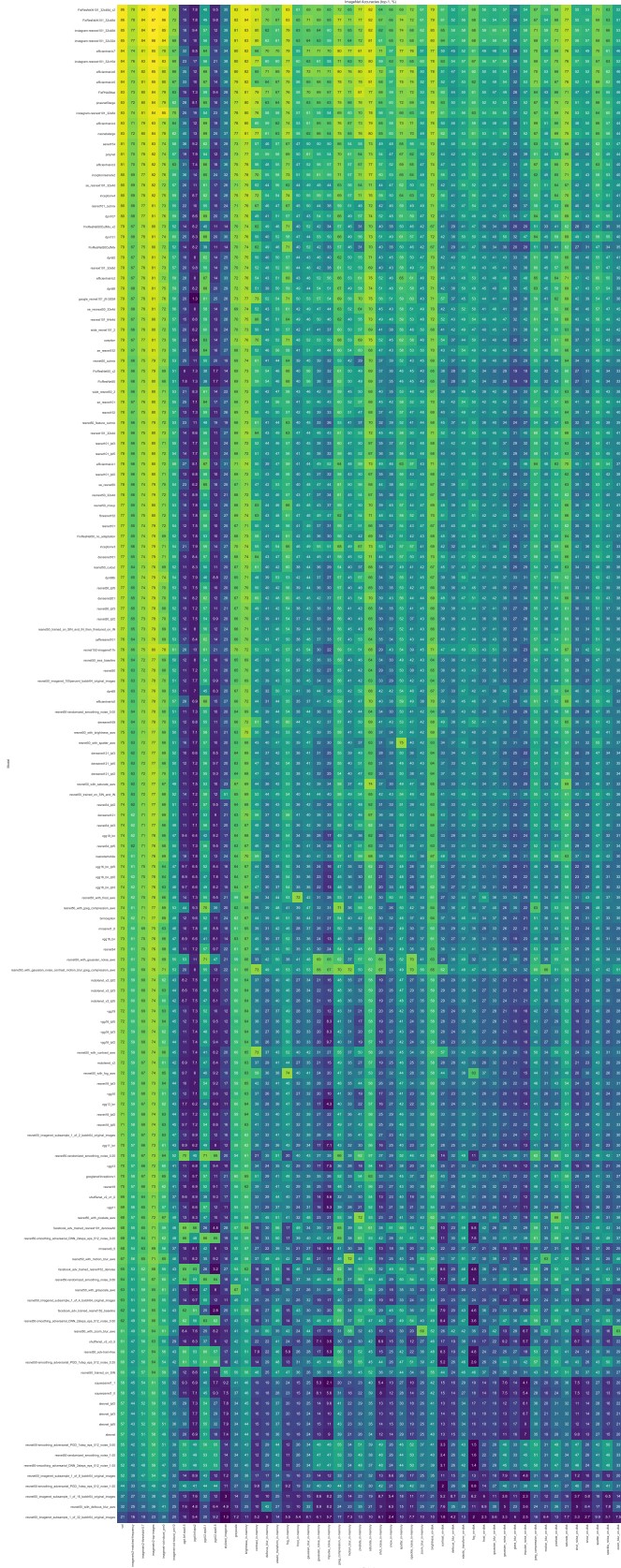

Figure 5: An overview of our testbed. Each row is a model, and each column is an evaluation setting. For the corruptions, we average over the five severities defined in (Hendrycks & Dietterich, 2019). We also plot in-memory and on-disk versions of each corruption as jpeg compression was found to be a confounding factor in (Ford et al., 2019). We leave out the class-subsampled models and evaluations described in Section 6 for brevity.

A.1    SYNTHETIC CORRUPTIONS VS. EFFECTIVE ROBUSTNESS (IMAGENET-VID-ROBUST)

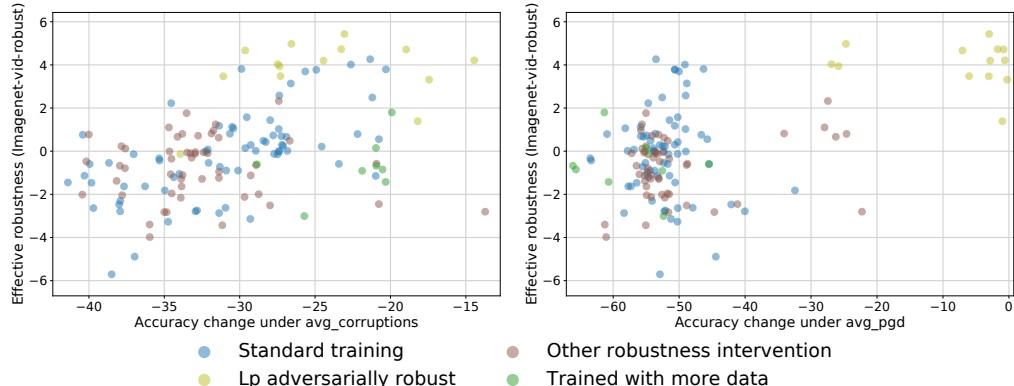

Figure 6: Robustness to synthetic distribution shift (x-axis) vs. effective robustness to the ImageNetVid distribution shift (y-axis). The left plot shows synthetic robustness as measured by an average over image corruptions from (Hendrycks & Dietterich, 2019; Geirhos et al., 2018; 2019). The right plot shows synthetic robustness as measured by PGD attacks for $\ell_2$- and $\ell_\infty$-robustness and two perturbation sizes each. In both cases, there is almost no correlation, with the except of the lp adversarially robust models explored earlier.

## A.2 SYNTHETIC ROBUSTNESS VS. EFFECTIVE NATURAL ROBUSTNESS

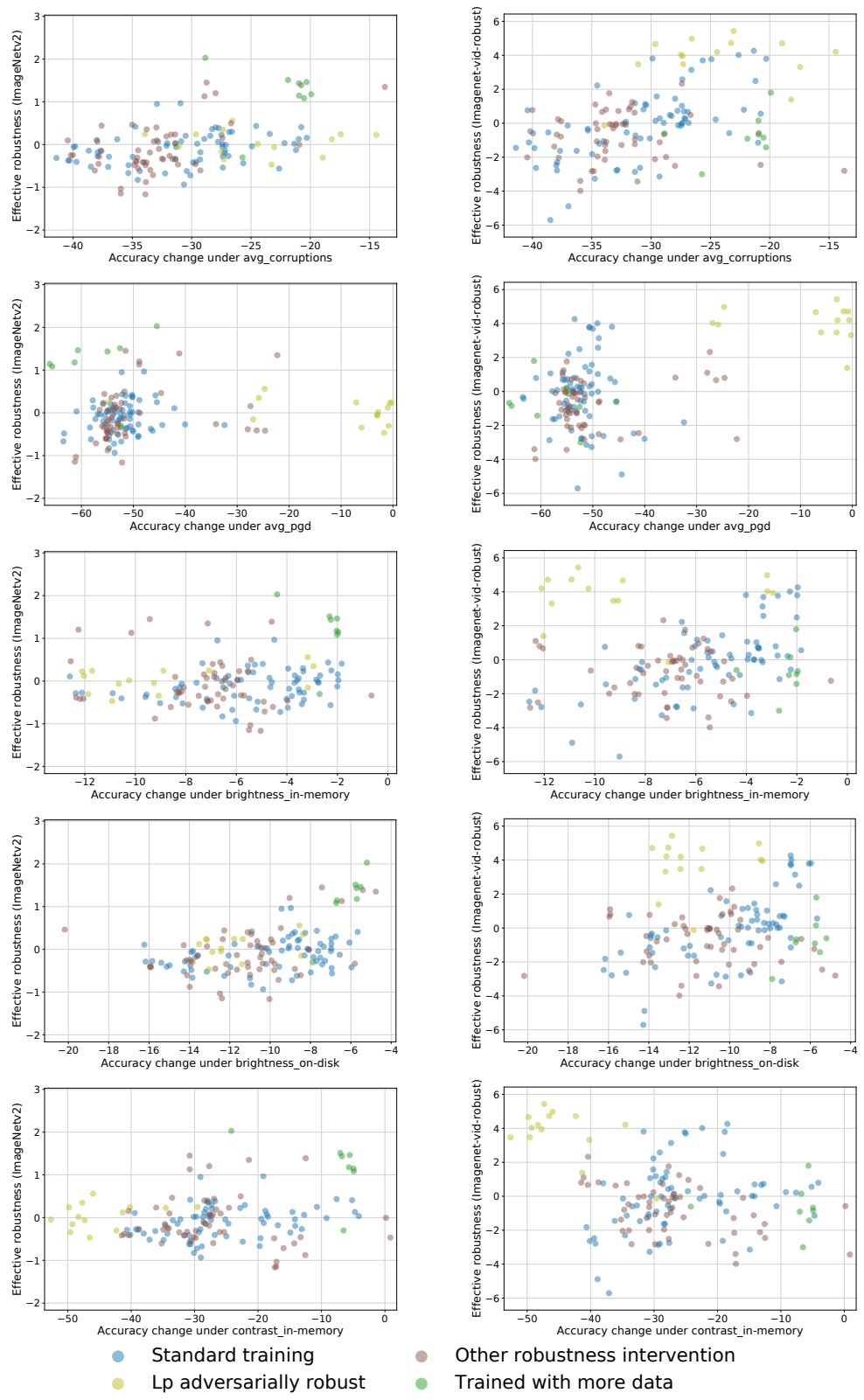

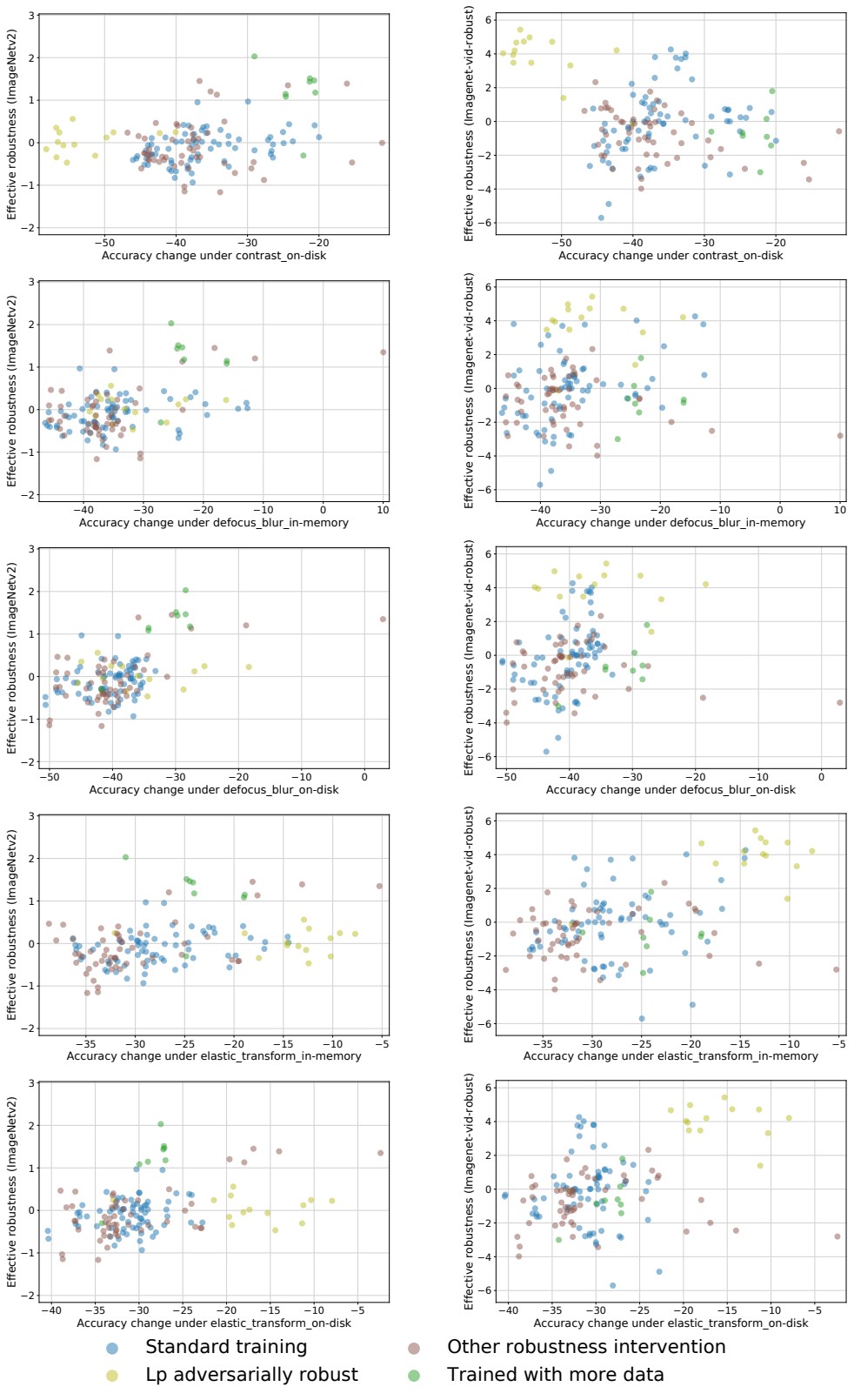

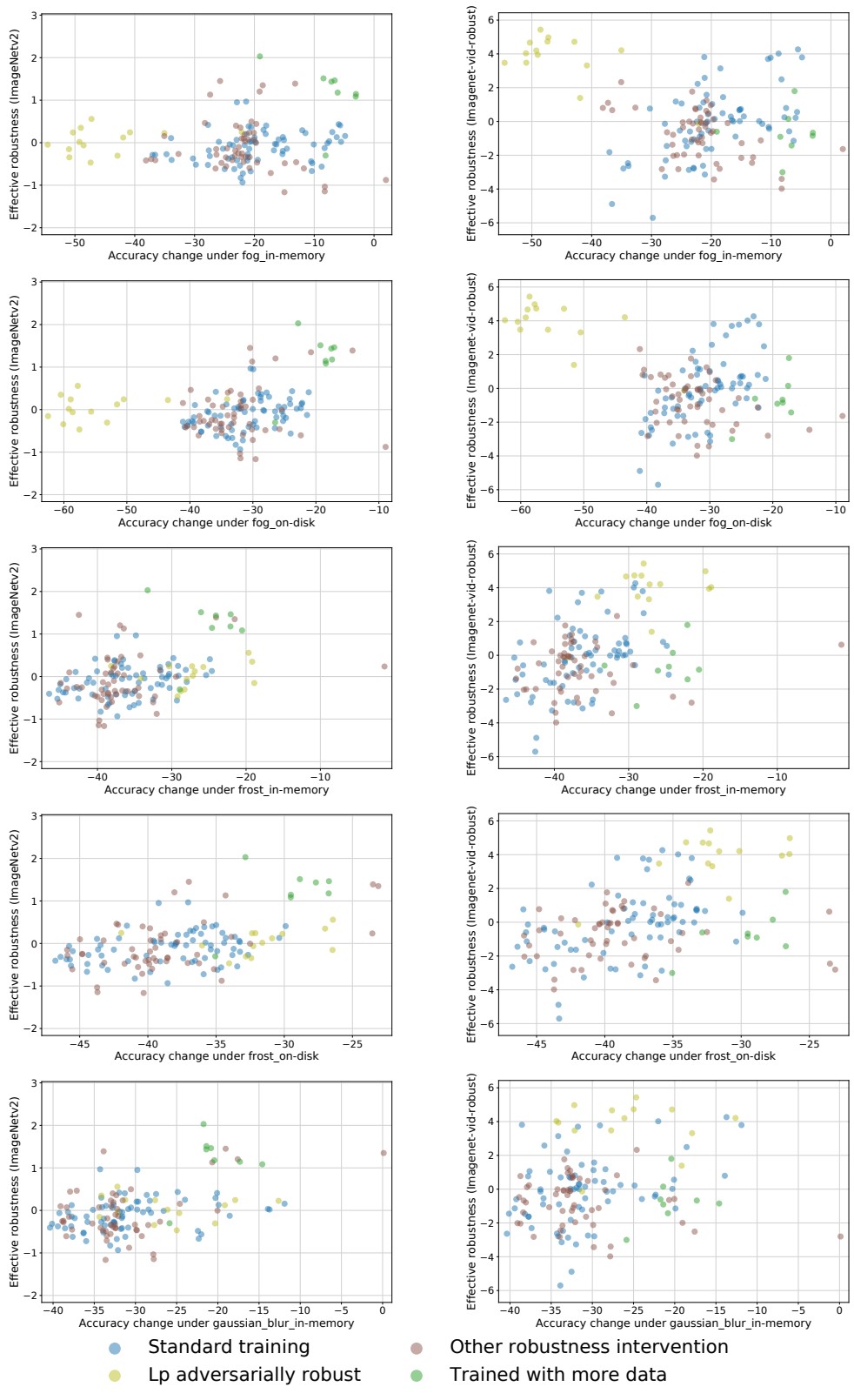

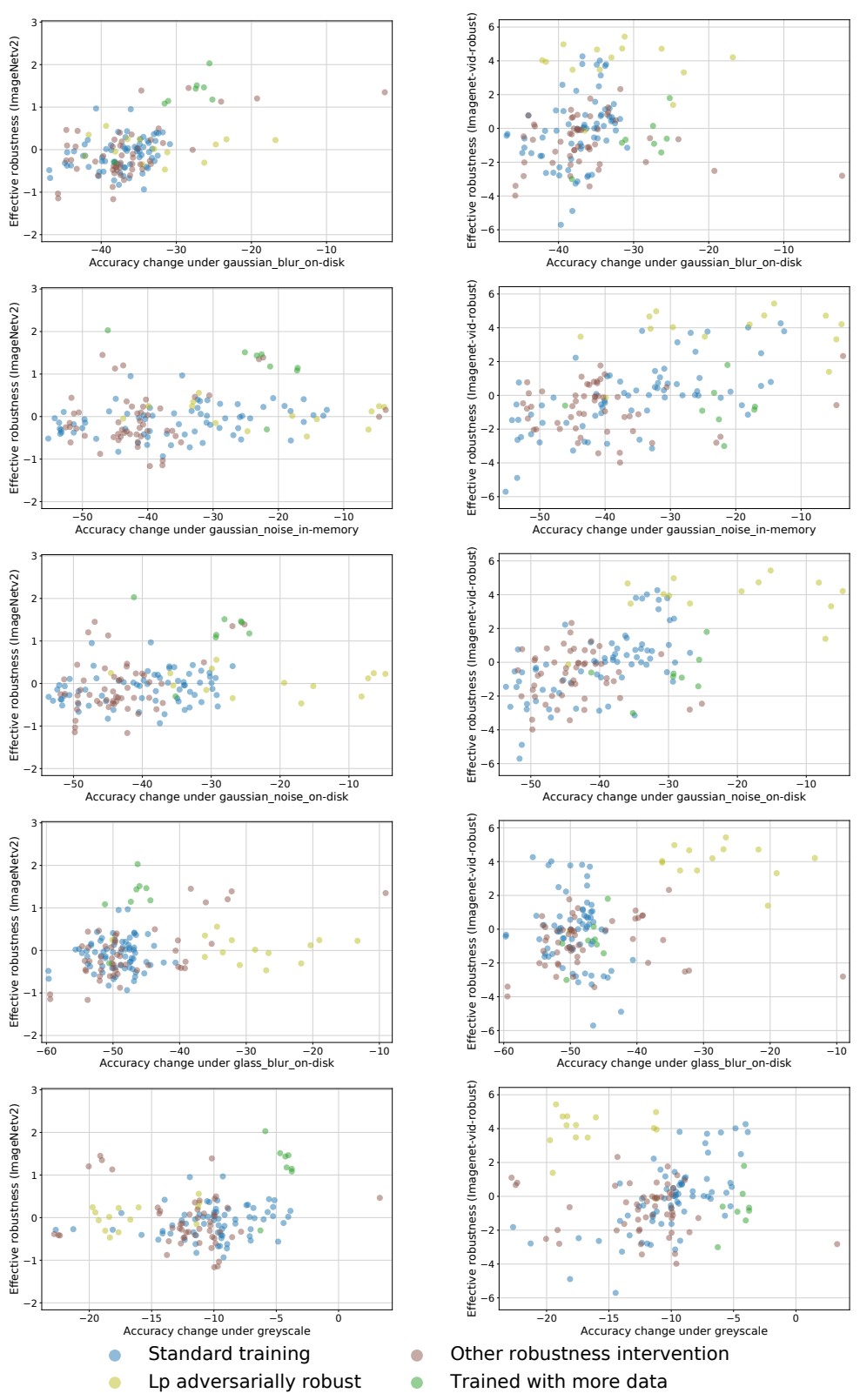

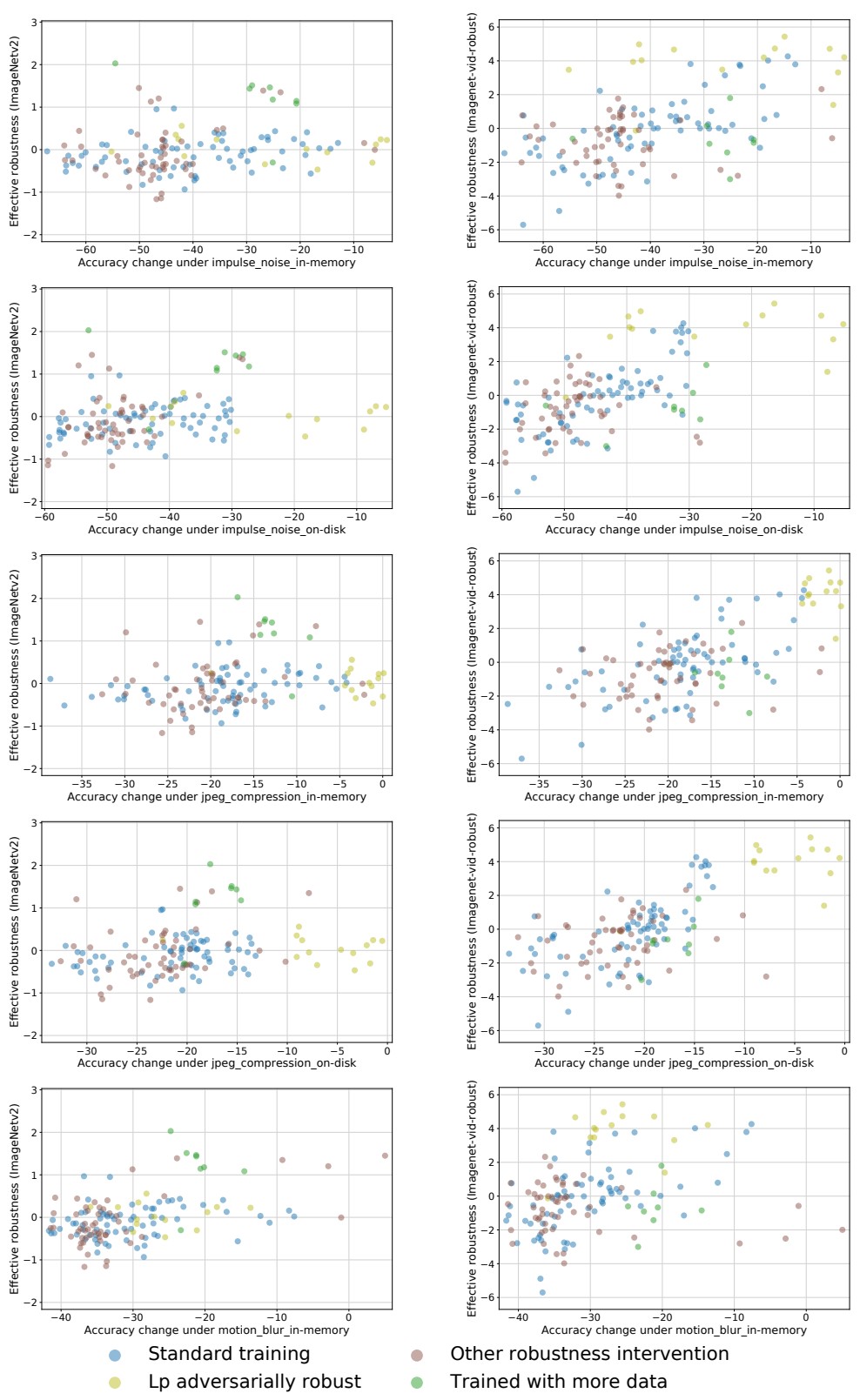

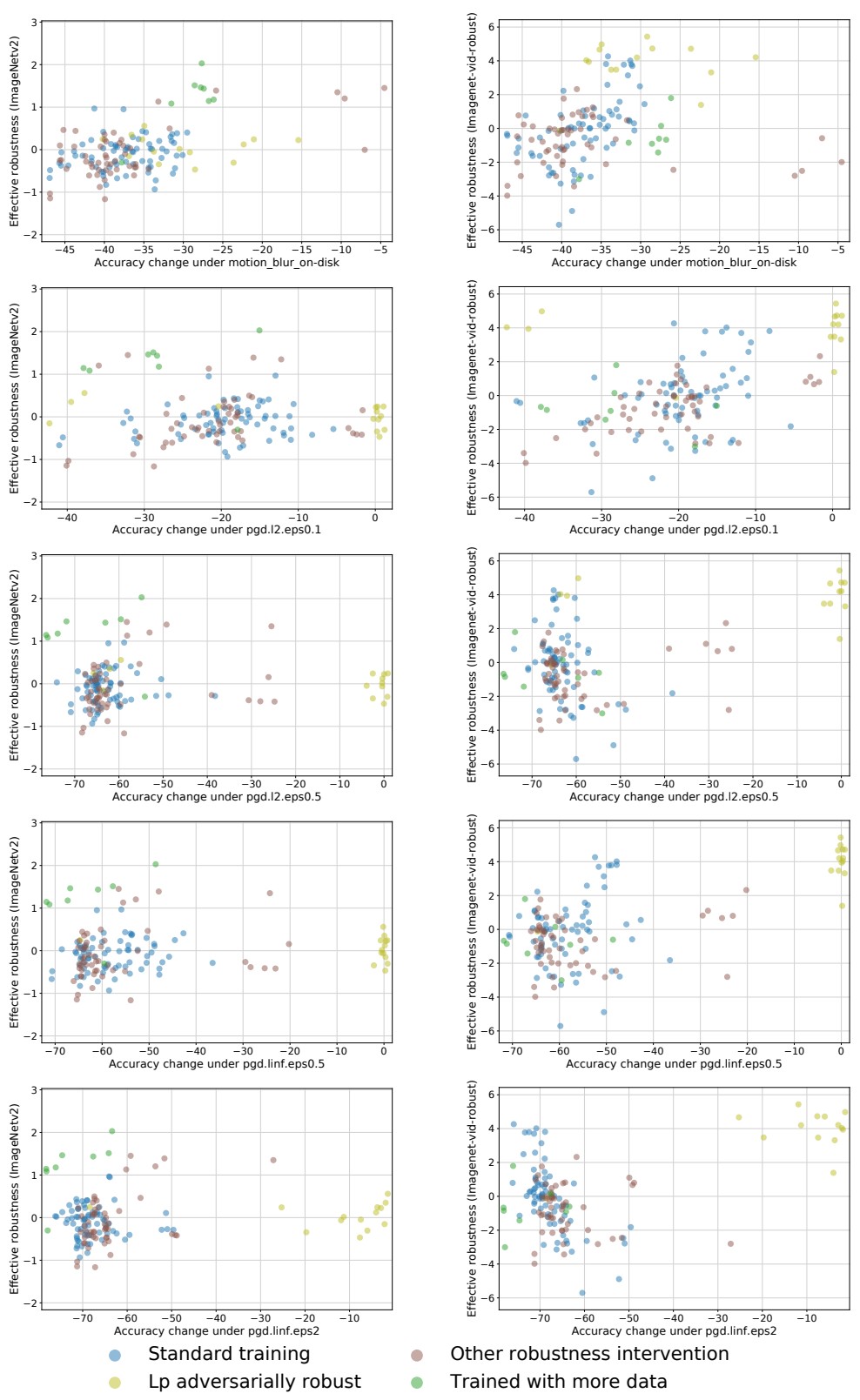

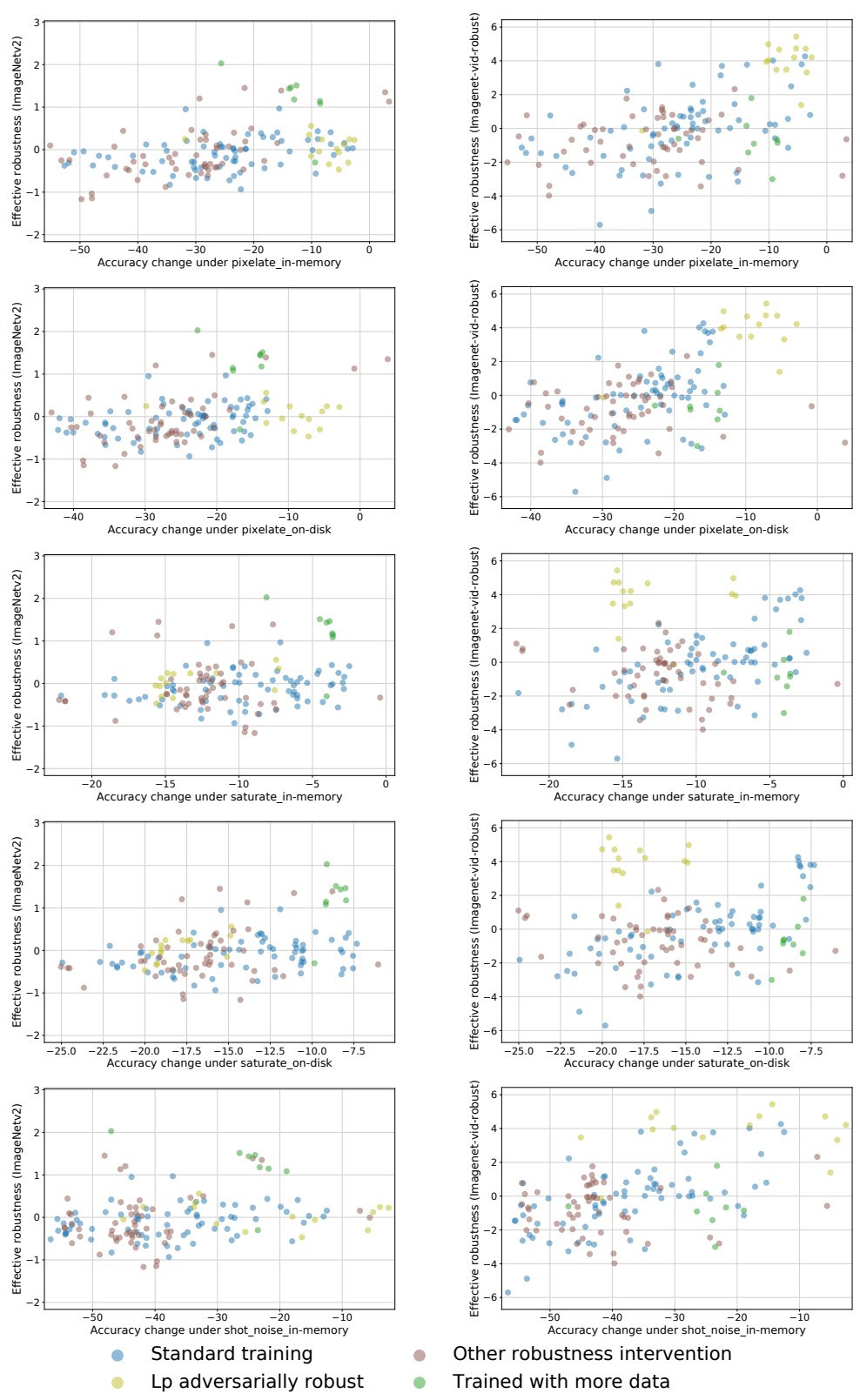

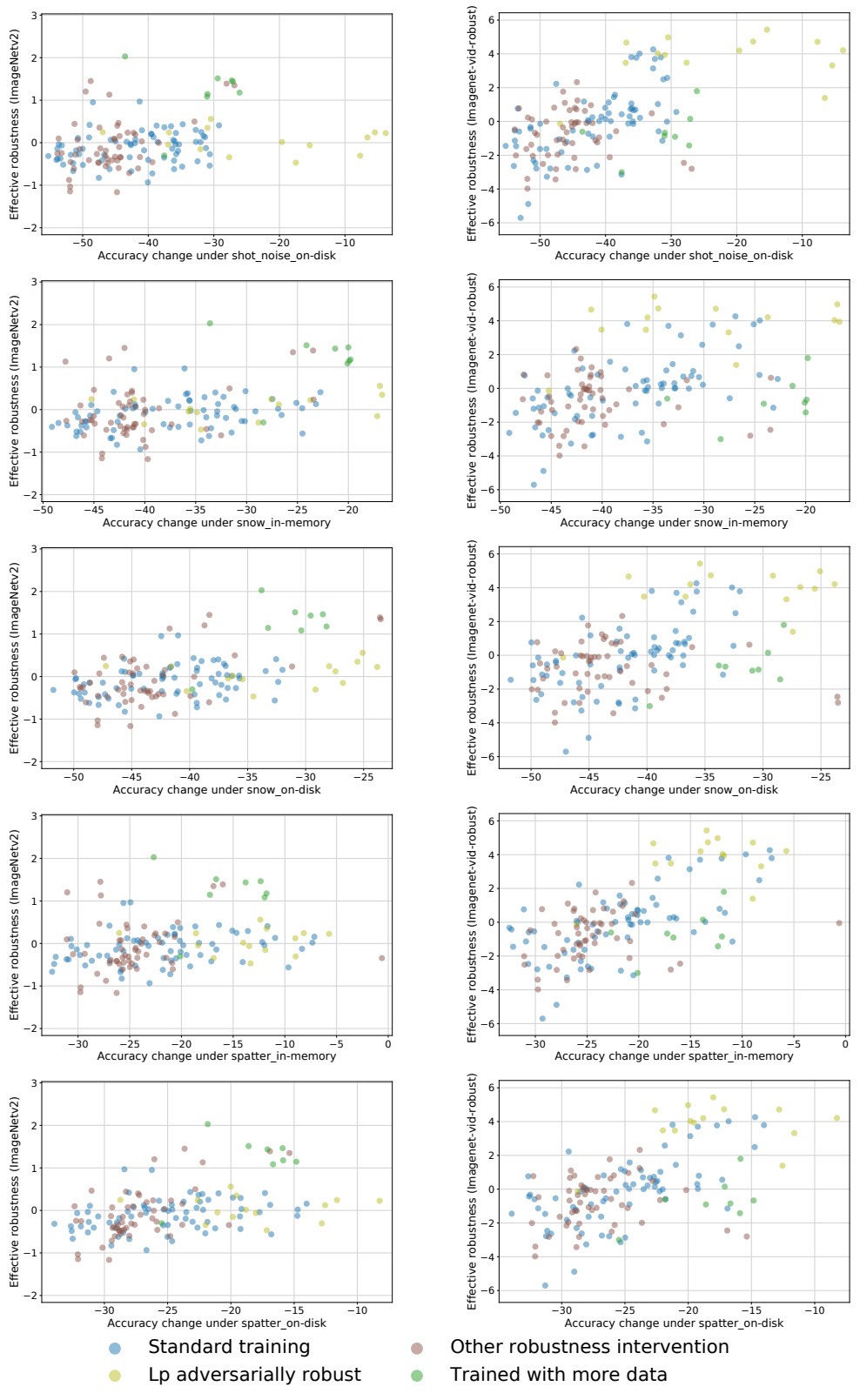

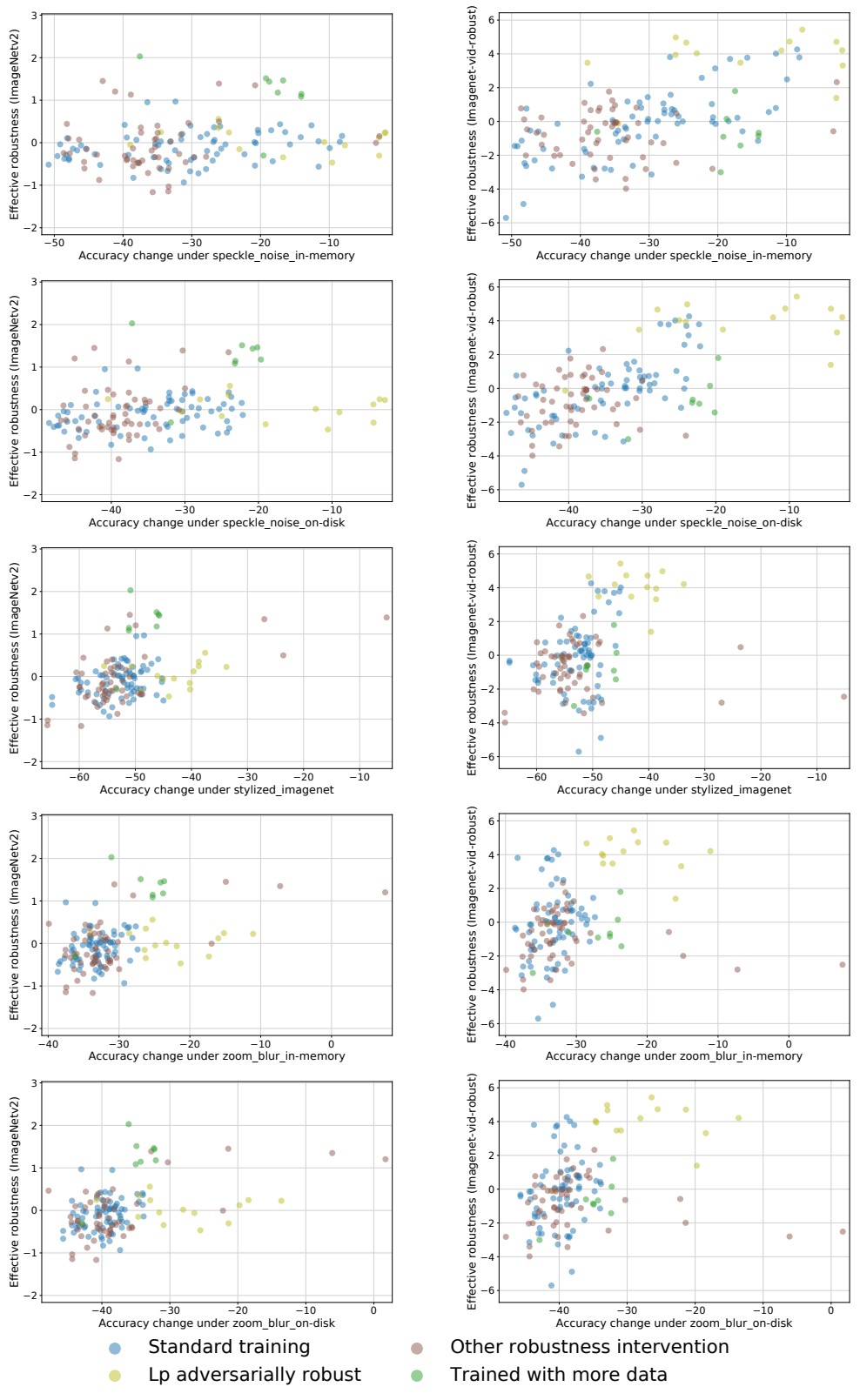

## A.3 Original accuracies vs. distribution shift accuracies

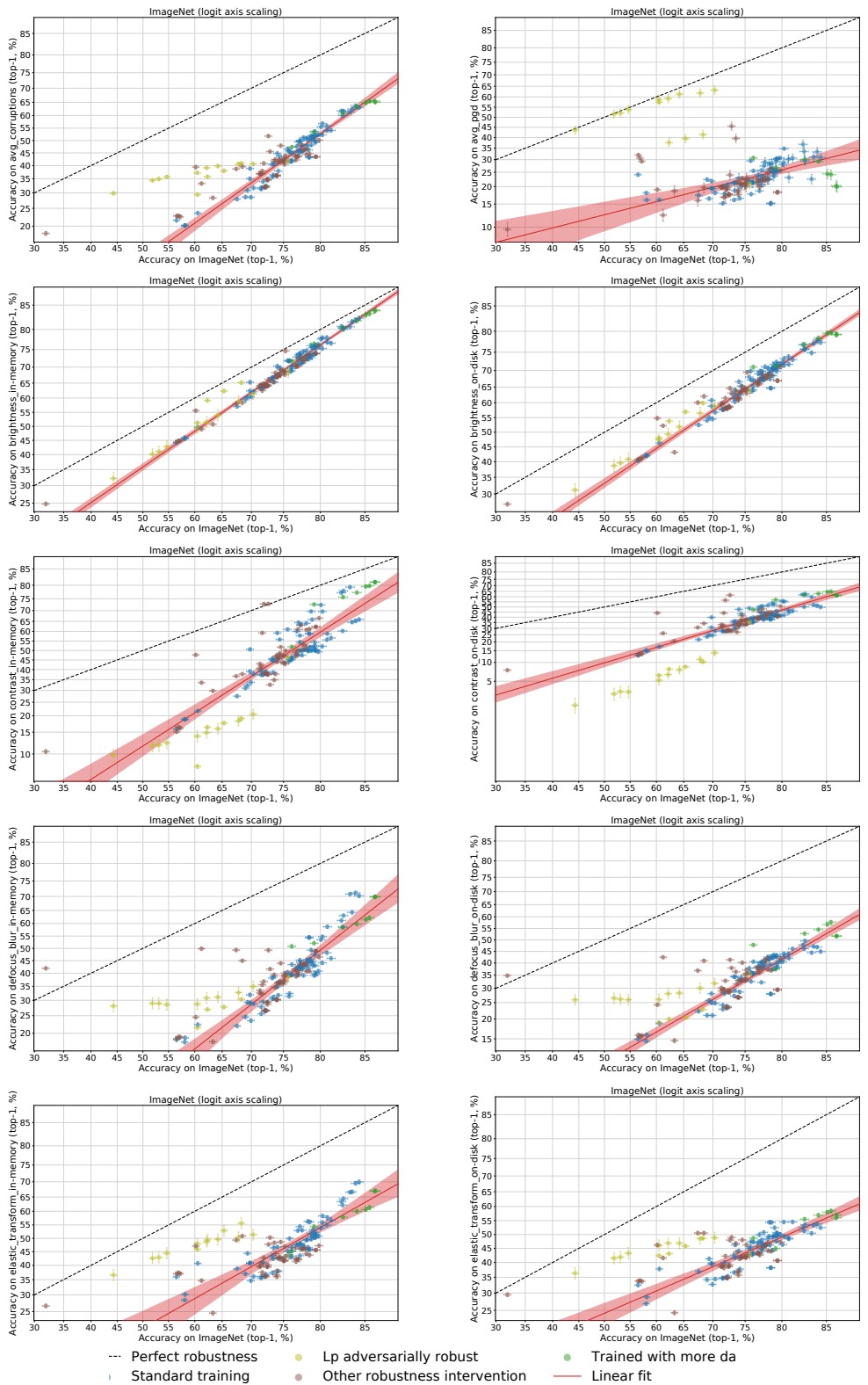

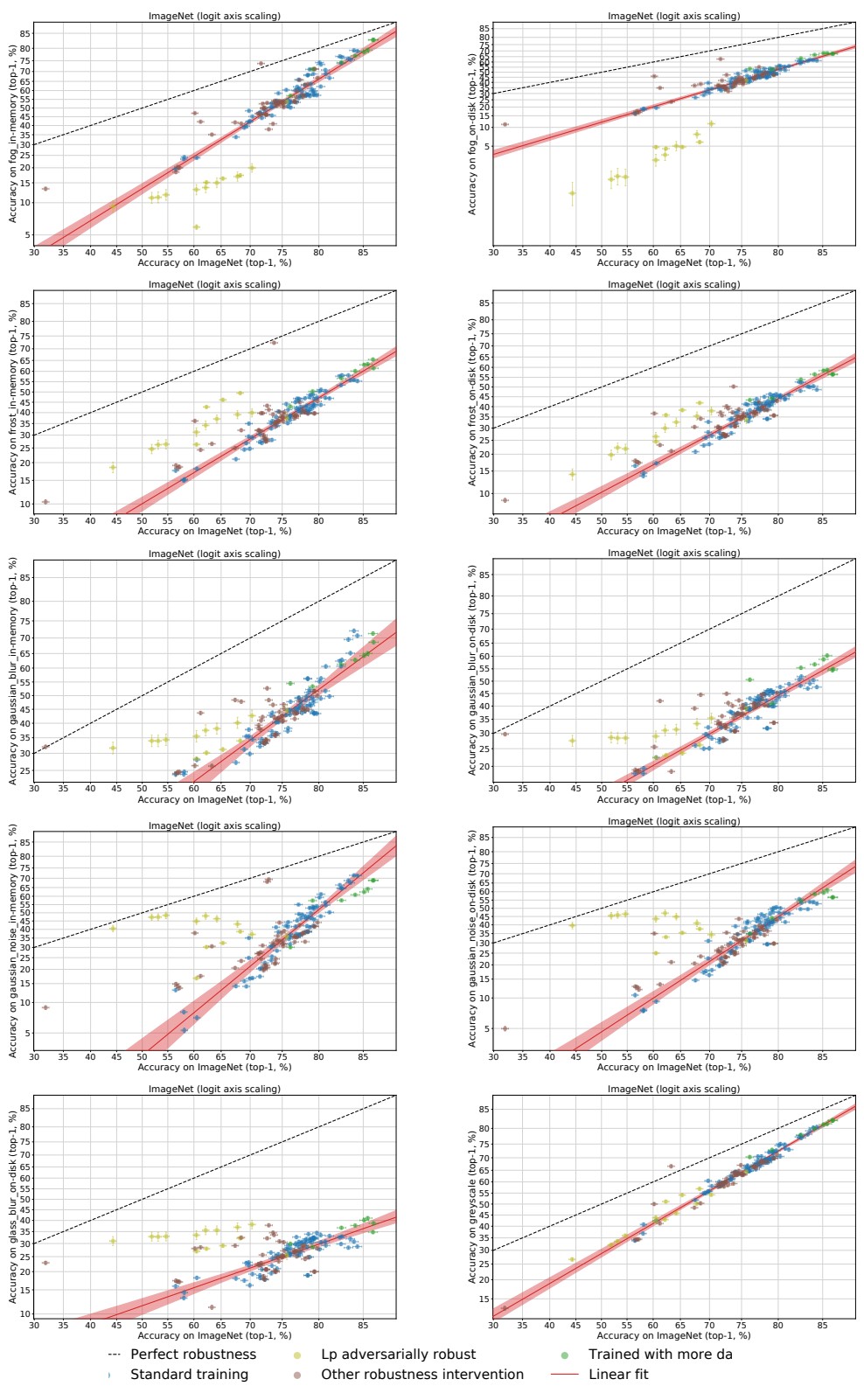

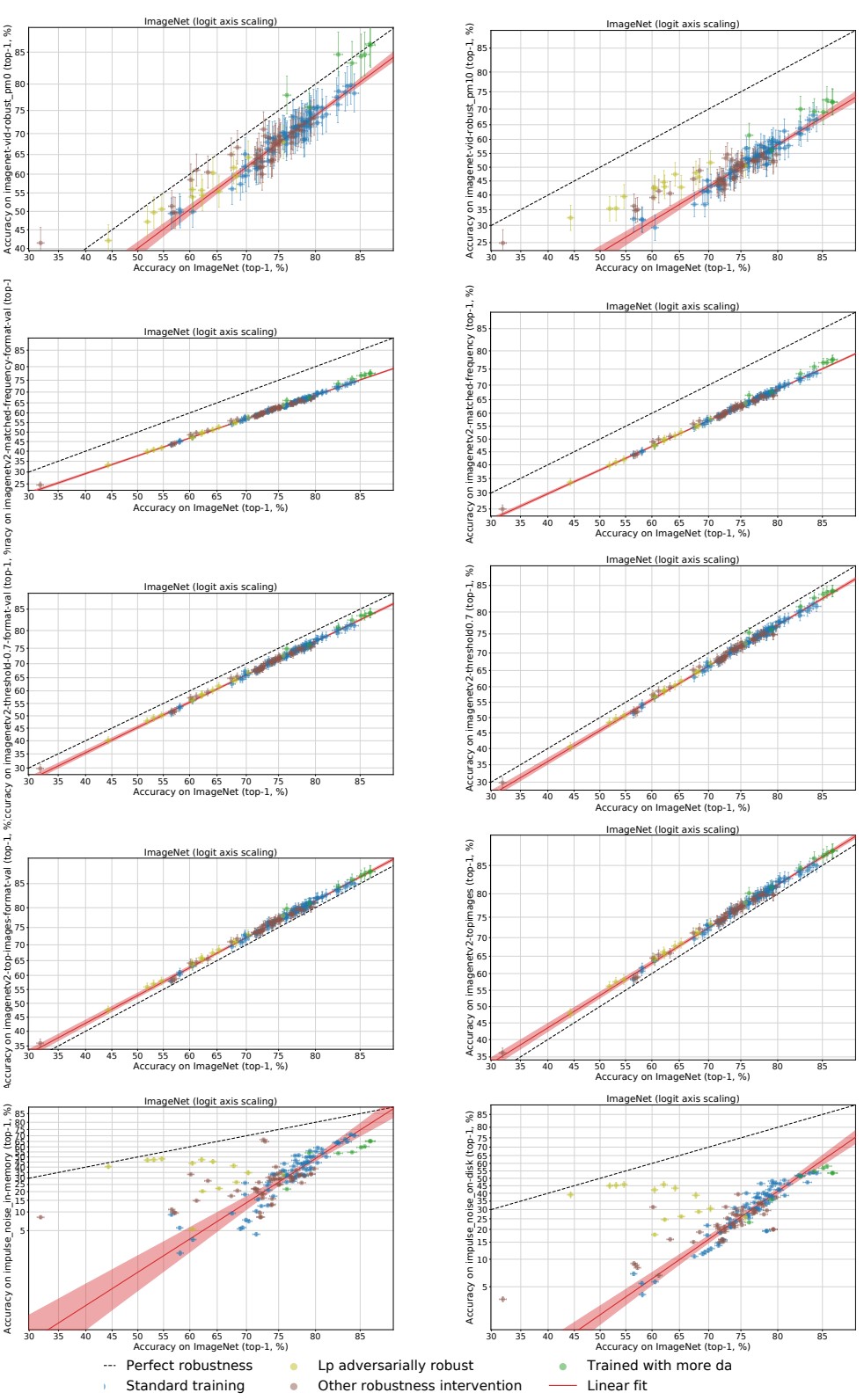

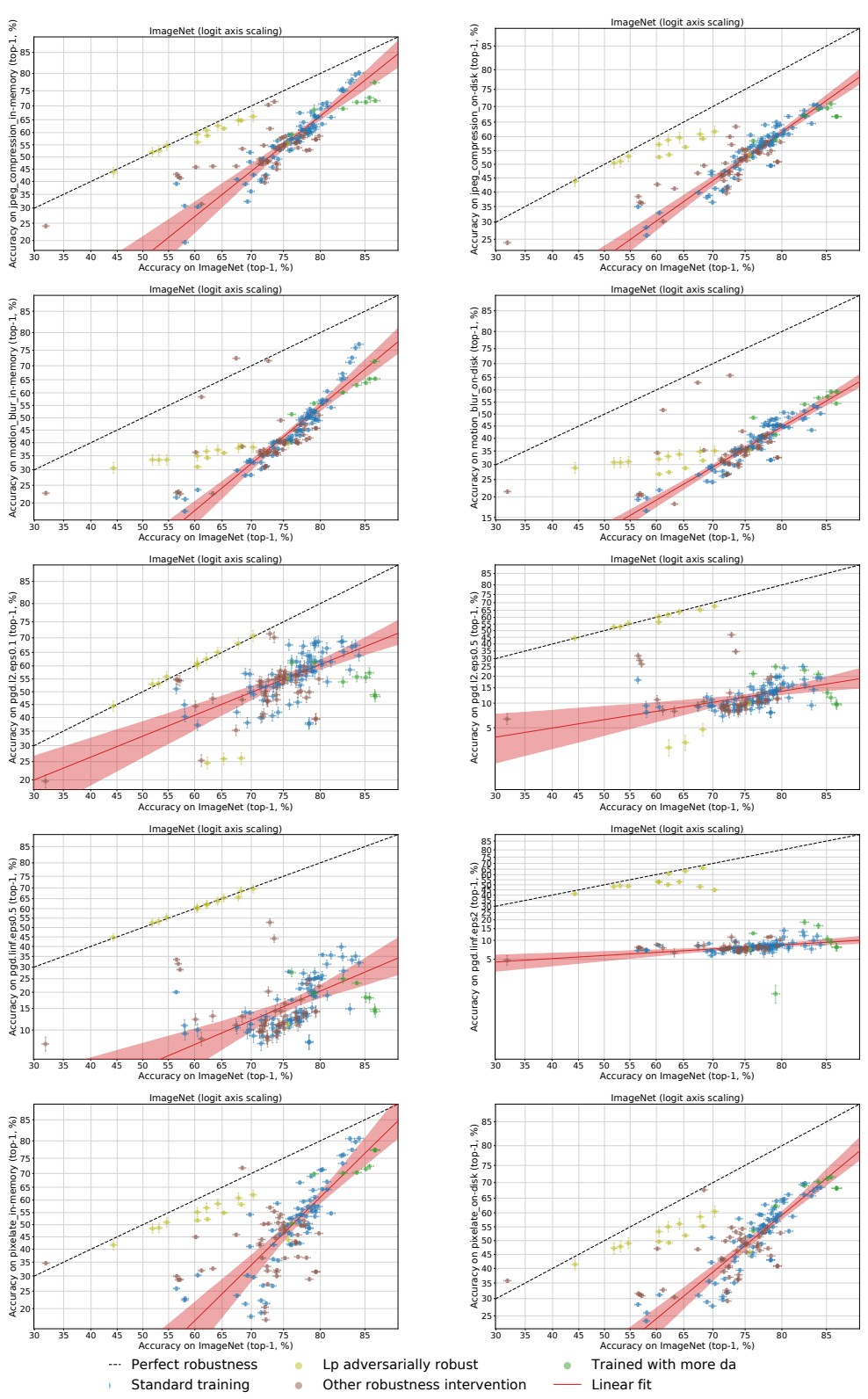

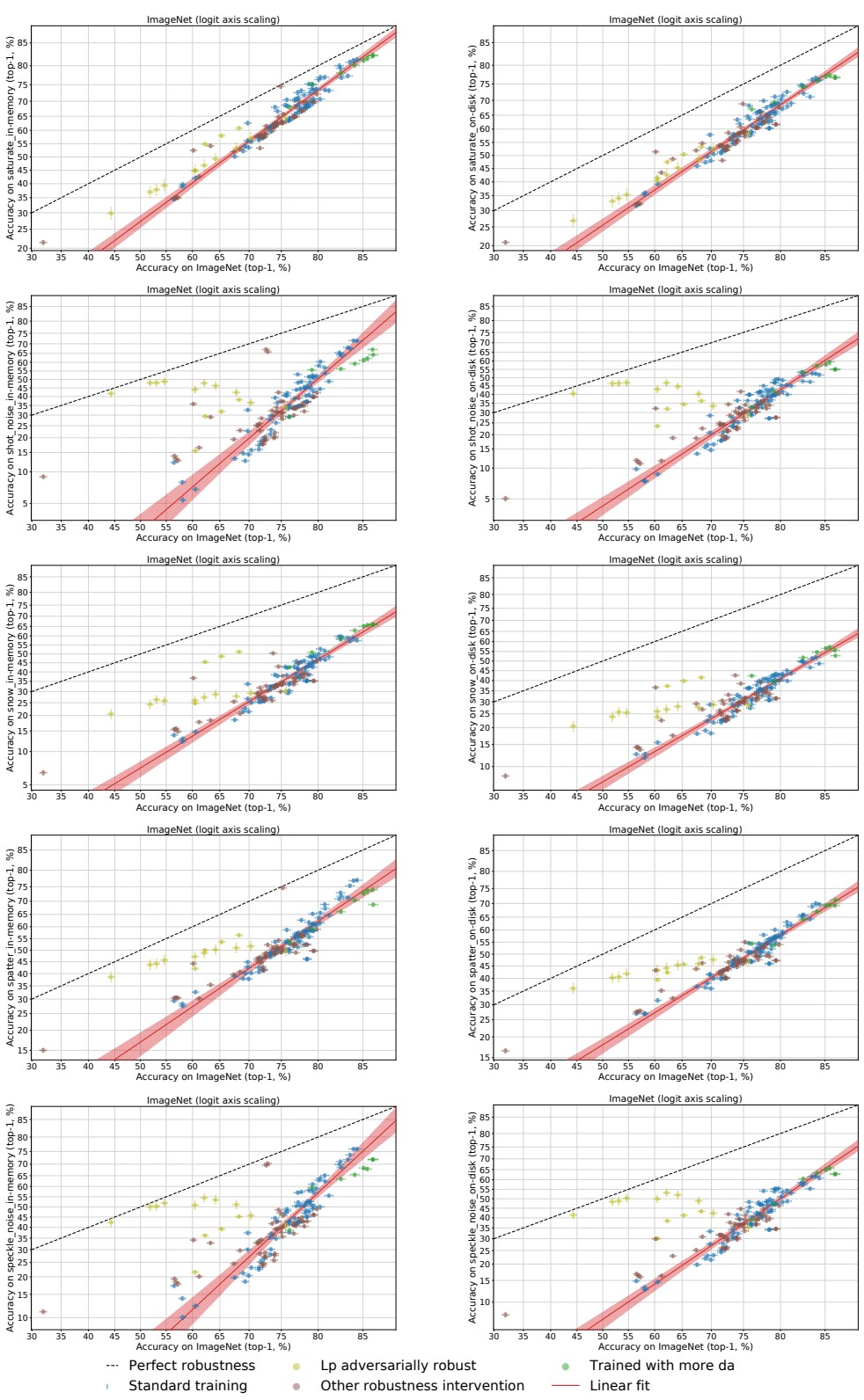

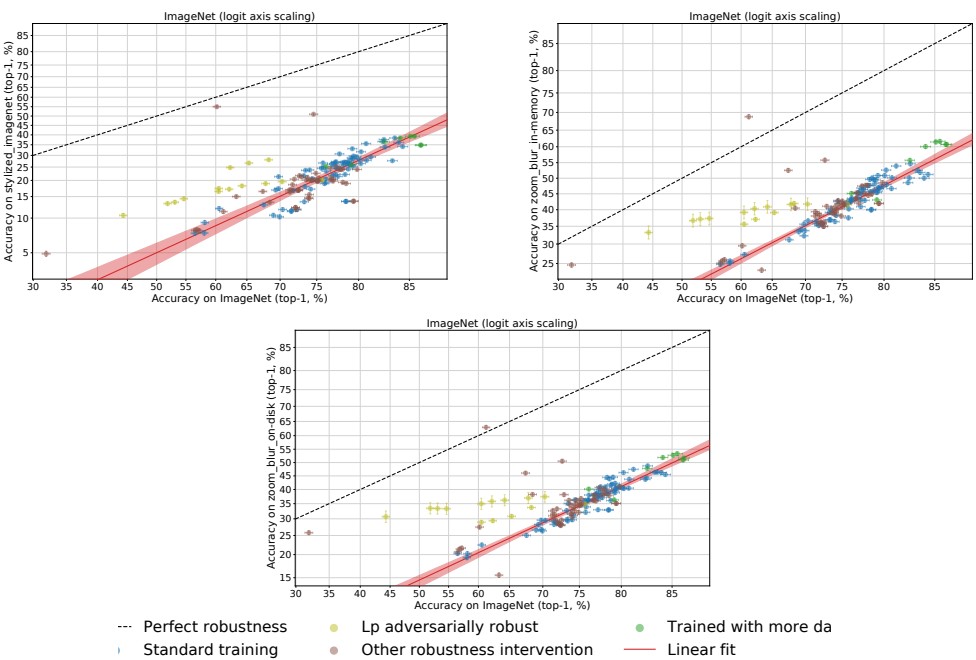

## A.4 CORRUPTION ROBUSTNESS

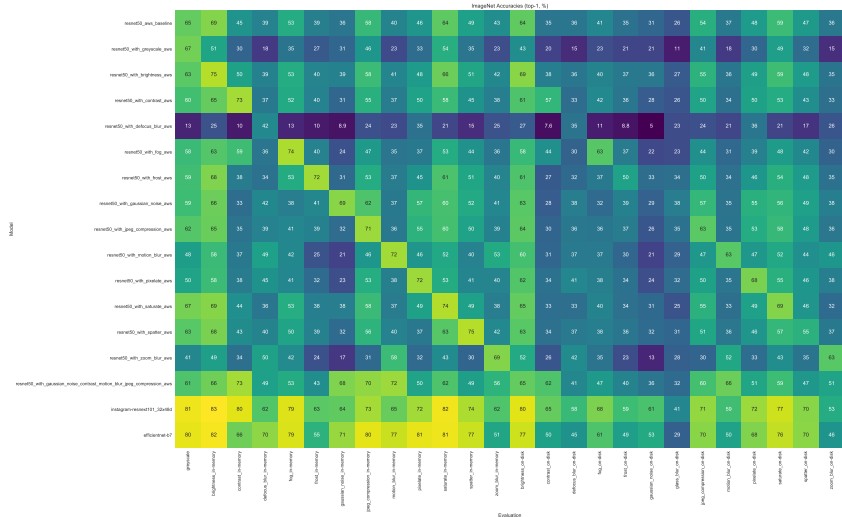

Figure 22: A detailed view of corruption robustness, with cells sampled from the main grid in Figure 5. Here we present resnet50s trained on some of the corruptions from the imagenet-c benchmark, as well as the best model trained on more data, instagram-resnext101_32x48d, and the best model trained on just the standard training set, efficientnet-b7.

We have already seen that corruption robustness does not promote effective robustness, or robustness to real distribution shift. Here, we analyze whether robustness to some corruptions transfers to others, and what may contribute to corruption robustness. Figure 22 shows the result of training various resnet50s[2] on a few corruptions from imagenet-c.

In line with prior work, this plot here tells us that training against one type of synthetic corruption or one set of synthetic corruption does not transfer well to other corruptions. There are cases where transfer does happen, but overall the models are only robust to the corruption they are trained on. This fact is even more prominent when looking at the difference between the instagram model and efficientnet - while both are amongst the top in terms of robust to synthetic corruptions, instagram is more robust to the *on disk* versions while efficientnet is more robust to the *in memory* versions. At first glance, this phenomenon seems strange, but in fact it could be that since the instagram model is trained on images pulled from instagram, which are jpeg compressed, it is more resistant to on disk perturbations - while the efficientnet is trained using autoaugment, which is an augmentation strategy done in memory, explaining its greater robustness to in memory perturbations.

## A.5 DETAILS ON OUR MODELS

---

[2]Each resnet50 was trained with a batch size of 256 for 120 epochs, starting with a learning rate of 0.1 and decaying by a factor of 10 every 30 epochs. For the resnet50s trained on corruptions, we randomly sample a corruption and severity for each image. Refer to A.6.3 for details on corruptions and severities. We use our custom fast gpu implementations of these corruptions for training.

Table 1: Top-1 model accuracies on the original ImageNet validation set, the Imagenetv2 test set (matched-frequency), an average over all the corruptions, and an average over all the pgd attacks. Note that since we take an average of many attacks, the PGD column can no longer be considered a worst-case attacker for the model (look to A.6.2 for specific attacks). The confidence intervals are 99.5% Clopper-Pearson intervals; they are not provided for the average columns since iid assumptions for the intervals may be violated. References for the models can be found in Appendix A.5.1. We exclude the subsampled class models for brevity. The table continues for 3 more pages.

| | ImageNet Aggregated Top-1 Accuracies | | | |
|---|---|---|---|---|
| Model | Validation Accuracy | Imagenetv2 Accuracy | Avg. Corr. Accuracy | Avg. PGD Accuracy |
| FixResNeXt101_32x48d_v2 | 86.0 [85.6, 86.4] | 77.7 [76.5, 78.9] | 65.1 | 19.9 |
| FixResNeXt101_32x48d | 85.9 [85.5, 86.4] | 77.6 [76.4, 78.7] | 65.4 | 20.3 |
| instagram-resnext101_32x48d | 85.4 [85.0, 85.9] | 77.0 [75.8, 78.1] | 65.5 | 24.1 |
| instagram-resnext101_32x32d | 85.1 [84.6, 85.5] | 76.8 [75.6, 77.9] | 64.8 | 24.4 |
| efficientnet-b7 | 84.4 [83.9, 84.9] | 74.4 [73.2, 75.6] | 63.1 | 30.9 |
| instagram-resnext101_32x16d | 84.2 [83.7, 84.6] | 75.5 [74.3, 76.7] | 63.2 | 29.2 |
| efficientnet-b6 | 84.0 [83.6, 84.5] | 74.1 [72.8, 75.3] | 63.7 | 33.4 |
| efficientnet-b5 | 83.7 [83.2, 84.1] | 73.3 [72.0, 74.5] | 62.5 | 31.1 |
| FixPNASNet | 83.5 [83.0, 83.9] | 73.2 [71.9, 74.4] | 61.3 | 22.5 |
| pnasnet5large | 82.7 [82.3, 83.2] | 72.3 [71.0, 73.6] | 61.8 | 29.5 |
| instagram-resnext101_32x8d | 82.7 [82.2, 83.2] | 73.6 [72.4, 74.9] | 60.8 | 30.1 |
| efficientnet-b4 | 82.6 [82.2, 83.1] | 71.5 [70.2, 72.7] | 60.0 | 33.5 |
| nasnetalarge | 82.5 [82.0, 83.0] | 72.3 [71.0, 73.5] | 61.7 | 36.8 |
| senet154 | 81.3 [80.8, 81.8] | 70.2 [68.9, 71.5] | 54.1 | 30.6 |
| polynet | 80.9 [80.4, 81.3] | 70.1 [68.8, 71.3] | 54.0 | 23.0 |
| efficientnet-b3 | 80.8 [80.3, 81.3] | 69.7 [68.4, 71.0] | 55.9 | 30.2 |
| inceptionresnetv2 | 80.3 [79.8, 80.8] | 69.3 [68.0, 70.6] | 56.9 | 34.8 |
| se_resnext101_32x4d | 80.2 [79.7, 80.7] | 69.4 [68.1, 70.7] | 52.3 | 28.8 |
| inceptionv4 | 80.1 [79.6, 80.6] | 69.2 [67.9, 70.5] | 55.5 | 27.9 |
| resnet101_cutmix | 79.8 [79.3, 80.3] | 68.0 [66.7, 69.3] | 50.1 | 25.6 |
| dpn107 | 79.7 [79.2, 80.2] | 67.8 [66.5, 69.2] | 52.4 | 30.8 |
| FixResNet50CutMix_v2 | 79.4 [78.9, 80.0] | 67.0 [65.7, 68.3] | 43.5 | 18.4 |
| dpn131 | 79.4 [78.9, 79.9] | 67.7 [66.4, 69.0] | 52.1 | 30.4 |
| FixResNet50CutMix | 79.4 [78.9, 79.9] | 66.8 [65.5, 68.2] | 43.4 | 18.2 |
| dpn92 | 79.4 [78.9, 79.9] | 67.2 [65.9, 68.6] | 49.3 | 25.7 |
| resnext101_32x8d | 79.3 [78.8, 79.8] | 67.4 [66.1, 68.8] | 49.7 | 25.4 |
| efficientnet-b2 | 79.3 [78.8, 79.8] | 67.4 [66.1, 68.7] | 53.6 | 29.3 |
| dpn98 | 79.2 [78.7, 79.7] | 67.8 [66.4, 69.1] | 51.8 | 30.2 |
| google_resnet101_jft-300M | 79.2 [78.7, 79.7] | 67.4 [66.1, 68.7] | 53.5 | 26.8 |
| se_resnext50_32x4d | 79.1 [78.6, 79.6] | 67.8 [66.4, 69.1] | 50.6 | 24.7 |
| resnext101_64x4d | 79.0 [78.4, 79.5] | 67.1 [65.8, 68.5] | 52.1 | 23.6 |
| wide_resnet101_2 | 78.8 [78.3, 79.4] | 66.4 [65.0, 67.7] | 48.2 | 25.2 |
| xception | 78.8 [78.3, 79.3] | 67.2 [65.9, 68.6] | 51.7 | 26.3 |
| se_resnet152 | 78.7 [78.1, 79.2] | 67.4 [66.1, 68.8] | 50.9 | 28.4 |
| resnet50_cutmix | 78.6 [78.1, 79.1] | 65.8 [64.5, 67.1] | 44.7 | 26.4 |
| FixResNet50_v2 | 78.6 [78.1, 79.1] | 66.5 [65.2, 67.8] | 43.3 | 15.3 |
| FixResNet50 | 78.5 [78.0, 79.1] | 66.2 [64.9, 67.6] | 43.2 | 15.1 |
| wide_resnet50_2 | 78.5 [77.9, 79.0] | 66.2 [64.8, 67.5] | 46.2 | 26.1 |
| se_resnet101 | 78.4 [77.9, 78.9] | 67.1 [65.8, 68.4] | 50.1 | 28.2 |
| resnet152 | 78.3 [77.8, 78.8] | 66.8 [65.5, 68.2] | 47.8 | 22.5 |
| resnet50_feature_cutmix | 78.2 [77.7, 78.7] | 66.0 [64.7, 67.4] | 44.3 | 25.1 |
| resnext101_32x4d | 78.2 [77.7, 78.7] | 66.3 [64.9, 67.6] | 51.0 | 22.4 |
| resnet101_lpf3 | 78.1 [77.6, 78.6] | 66.1 [64.7, 67.4] | 46.5 | 22.5 |
| resnet101_lpf5 | 77.9 [77.4, 78.4] | 66.2 [64.8, 67.5] | 46.5 | 23.1 |
| efficientnet-b1 | 77.8 [77.3, 78.3] | 66.3 [65.0, 67.7] | 51.2 | 29.0 |
| resnet101_lpf2 | 77.8 [77.3, 78.3] | 66.3 [64.9, 67.6] | 46.1 | 22.0 |

| ImageNet Aggregated Top-1 Accuracies | | | | |
|---|---|---|---|---|
| Model | Validation Accuracy | Imagenetv2 Accuracy | Avg. Corr. Accuracy | Avg. PGD Accuracy |
| se_resnet50 | 77.6 [77.1, 78.2] | 65.8 [64.5, 67.2] | 48.1 | 27.4 |
| resnext50_32x4d | 77.6 [77.1, 78.1] | 65.8 [64.5, 67.2] | 45.6 | 22.5 |
| resnet50_mixup | 77.5 [76.9, 78.0] | 65.0 [63.7, 66.4] | 48.2 | 22.2 |
| fbresnet152 | 77.4 [76.9, 77.9] | 65.5 [64.1, 66.8] | 50.0 | 23.5 |
| resnet101 | 77.4 [76.8, 77.9] | 65.5 [64.1, 66.8] | 46.1 | 21.8 |
| FixResNet50_no_adaptation | 77.3 [76.8, 77.9] | 65.4 [64.1, 66.8] | 45.2 | 20.5 |
| inceptionv3 | 77.3 [76.8, 77.8] | 65.7 [64.3, 67.0] | 49.8 | 25.7 |
| densenet161 | 77.1 [76.6, 77.7] | 65.3 [63.9, 66.6] | 49.4 | 22.2 |
| resnet50_cutout | 77.1 [76.5, 77.6] | 64.4 [63.1, 65.8] | 43.8 | 19.9 |
| dpn68b | 77.0 [76.5, 77.6] | 64.8 [63.4, 66.1] | 45.7 | 18.7 |
| resnet50_lpf5 | 77.0 [76.5, 77.6] | 64.5 [63.1, 65.8] | 43.5 | 21.7 |
| densenet201 | 76.9 [76.4, 77.4] | 64.8 [63.4, 66.1] | 47.6 | 23.9 |
| resnet50_lpf3 | 76.8 [76.3, 77.3] | 64.7 [63.3, 66.0] | 43.3 | 21.8 |
| resnet50_lpf2 | 76.8 [76.3, 77.3] | 64.5 [63.2, 65.8] | 42.2 | 20.8 |
| resnet50_trained_on_SIN_a nd_IN_then_finetuned_on_IN | 76.7 [76.2, 77.2] | 64.6 [63.3, 66.0] | 44.0 | 22.6 |
| cafferesnet101 | 76.2 [75.7, 76.7] | 64.1 [62.8, 65.5] | 44.8 | 25.5 |
| resnet152-imagenet11k | 76.2 [75.6, 76.7] | 66.1 [64.8, 67.5] | 47.3 | 30.7 |
| resnet50_aws_baseline | 76.1 [75.6, 76.7] | 63.6 [62.3, 65.0] | 42.1 | 21.3 |
| resnet50 | 76.1 [75.6, 76.7] | 63.2 [61.8, 64.6] | 41.6 | 21.4 |
| resnet50_imagenet_100perce nt_batch64_original_images | 76.0 [75.4, 76.5] | 63.2 [61.9, 64.6] | 41.6 | 21.3 |
| dpn68 | 75.9 [75.3, 76.4] | 63.1 [61.7, 64.5] | 45.5 | 17.7 |
| efficientnet-b0 | 75.8 [75.3, 76.3] | 63.2 [61.9, 64.6] | 45.9 | 29.5 |
| resnet50-randomized_ smoothing_noise_0.00 | 75.7 [75.1, 76.2] | 63.8 [62.4, 65.1] | 41.8 | 20.9 |
| densenet169 | 75.6 [75.1, 76.1] | 63.5 [62.2, 64.9] | 46.7 | 21.8 |
| resnet50_with_brightness_aws | 75.3 [74.7, 75.8] | 62.7 [61.3, 64.1] | 43.9 | 22.2 |
| resnet50_with_spatter_aws | 75.2 [74.7, 75.8] | 62.6 [61.3, 64.0] | 42.8 | 22.8 |
| densenet121_lpf3 | 75.1 [74.6, 75.7] | 62.5 [61.1, 63.8] | 40.5 | 20.0 |
| densenet121_lpf5 | 75.0 [74.5, 75.6] | 62.8 [61.5, 64.2] | 41.8 | 21.1 |
| densenet121_lpf2 | 75.0 [74.5, 75.6] | 63.1 [61.7, 64.5] | 41.2 | 20.8 |
| resnet50_with_saturate_aws | 74.9 [74.3, 75.4] | 62.3 [60.9, 63.6] | 42.4 | 20.6 |
| resnet50_trained_on_SIN_and_IN | 74.6 [74.0, 75.1] | 62.8 [61.4, 64.1] | 47.9 | 22.9 |
| resnet34_lpf2 | 74.5 [73.9, 75.0] | 62.2 [60.8, 63.6] | 41.5 | 21.0 |
| densenet121 | 74.4 [73.9, 75.0] | 62.1 [60.8, 63.5] | 43.5 | 20.0 |
| resnet34_lpf3 | 74.3 [73.8, 74.9] | 62.2 [60.8, 63.5] | 42.2 | 20.7 |
| vgg19_bn | 74.2 [73.7, 74.8] | 62.0 [60.6, 63.3] | 37.9 | 16.5 |
| resnet34_lpf5 | 74.2 [73.6, 74.7] | 62.2 [60.8, 63.6] | 41.2 | 20.9 |
| nasnetamobile | 74.1 [73.5, 74.6] | 61.3 [59.9, 62.7] | 44.8 | 22.8 |
| vgg16_bn_lpf5 | 74.0 [73.5, 74.6] | 61.2 [59.8, 62.5] | 36.2 | 19.0 |
| vgg16_bn_lpf2 | 74.0 [73.5, 74.6] | 61.7 [60.3, 63.0] | 36.1 | 17.5 |
| vgg16_bn_lpf3 | 73.9 [73.4, 74.5] | 61.9 [60.6, 63.3] | 36.3 | 18.3 |
| resnet50_with_frost_aws | 73.8 [73.2, 74.3] | 61.6 [60.2, 62.9] | 42.4 | 21.5 |
| resnet50_with_jpeg_compression_aws | 73.6 [73.1, 74.2] | 60.9 [59.5, 62.3] | 41.8 | 39.5 |
| bninception | 73.5 [73.0, 74.1] | 62.0 [60.6, 63.4] | 40.6 | 21.3 |
| mnasnet1_0 | 73.5 [72.9, 74.0] | 60.4 [59.1, 61.8] | 36.4 | 18.8 |
| vgg16_bn | 73.4 [72.8, 73.9] | 60.7 [59.3, 62.1] | 35.7 | 16.1 |
| resnet34 | 73.3 [72.8, 73.9] | 60.9 [59.5, 62.2] | 40.5 | 21.2 |
| resnet50_with_gaussian_noise_aws | 73.0 [72.4, 73.5] | 60.6 [59.2, 61.9] | 45.6 | 45.5 |
| resnet50_with_gaussian_noise_contra st_motion_blur_jpeg_compression_aws | 72.7 [72.2, 73.3] | 60.1 [58.7, 61.5] | 51.8 | 24.0 |
| mobilenet_v2_lpf2 | 72.6 [72.1, 73.2] | 59.4 [58.0, 60.8] | 34.5 | 17.4 |
| mobilenet_v2_lpf3 | 72.6 [72.0, 73.1] | 59.7 [58.3, 61.1] | 34.8 | 17.6 |

| ImageNet Aggregated Top-1 Accuracies | | | | |
|---|---|---|---|---|
| Model | Validation Accuracy | Imagenetv2 Accuracy | Avg. Corr. Accuracy | Avg. PGD Accuracy |
| mobilenet_v2_lpf5 | 72.5 [71.9, 73.1] | 59.8 [58.4, 61.1] | 34.9 | 17.8 |
| vgg19 | 72.4 [71.8, 72.9] | 59.7 [58.3, 61.1] | 32.4 | 20.6 |
| vgg16_lpf5 | 72.3 [71.8, 72.9] | 59.8 [58.4, 61.2] | 31.9 | 19.7 |
| vgg16_lpf3 | 72.2 [71.6, 72.7] | 59.3 [57.9, 60.7] | 32.2 | 19.1 |
| vgg16_lpf2 | 72.2 [71.6, 72.7] | 59.3 [57.9, 60.6] | 32.0 | 19.2 |
| resnet50_with_contrast_aws | 72.0 [71.4, 72.6] | 58.9 [57.5, 60.3] | 40.8 | 17.0 |
| mobilenet_v2 | 71.9 [71.3, 72.4] | 59.0 [57.6, 60.4] | 34.0 | 17.8 |
| resnet50_with_fog_aws | 71.8 [71.2, 72.3] | 58.2 [56.8, 59.6] | 37.9 | 16.8 |
| resnet18_lpf3 | 71.7 [71.1, 72.2] | 58.5 [57.1, 59.9] | 36.8 | 20.1 |
| vgg16 | 71.6 [71.0, 72.2] | 58.5 [57.1, 59.9] | 31.3 | 20.1 |
| vgg13_bn | 71.6 [71.0, 72.2] | 58.8 [57.4, 60.2] | 31.8 | 15.2 |
| resnet18_lpf2 | 71.4 [70.8, 72.0] | 58.5 [57.1, 59.9] | 36.9 | 19.6 |
| resnet18_lpf5 | 71.4 [70.8, 72.0] | 58.1 [56.7, 59.5] | 36.9 | 19.9 |
| resnet50_imagenet_subsample_1 _of_2_batch64_original_images | 70.5 [69.9, 71.1] | 58.0 [56.6, 59.3] | 35.4 | 21.8 |
| vgg11_bn | 70.4 [69.8, 70.9] | 57.4 [56.0, 58.8] | 31.7 | 18.0 |
| resnet50-randomized_ smoothing_noise_0.25 | 70.3 [69.7, 70.9] | 57.7 [56.3, 59.1] | 40.7 | 63.2 |
| vgg13 | 69.9 [69.3, 70.5] | 56.8 [55.4, 58.2] | 28.5 | 19.3 |
| googlenet/inceptionv1 | 69.8 [69.2, 70.4] | 57.9 [56.5, 59.3] | 38.8 | 21.8 |
| resnet18 | 69.8 [69.2, 70.3] | 57.3 [55.9, 58.7] | 35.0 | 19.5 |
| shufflenet_v2_x1_0 | 69.4 [68.8, 69.9] | 56.0 [54.5, 57.3] | 30.9 | 16.4 |
| vgg11 | 69.0 [68.4, 69.6] | 55.7 [54.3, 57.1] | 28.6 | 22.4 |
| resnet50_with_pixelate_aws | 68.5 [67.9, 69.1] | 56.7 [55.3, 58.1] | 39.6 | 19.6 |
| facebook_adv_trained_ resnext101_denoiseAll | 68.3 [67.7, 68.9] | 55.2 [53.8, 56.6] | 40.9 | 41.4 |
| resnet50-smoothing_adversarial _DNN_2steps_eps_512_noise_0.25 | 67.9 [67.3, 68.5] | 54.5 [53.1, 55.9] | 40.6 | 61.8 |
| mnasnet0_5 | 67.6 [67.0, 68.2] | 54.2 [52.8, 55.6] | 27.9 | 17.4 |
| resnet50_with_motion_blur_aws | 67.5 [66.9, 68.0] | 55.9 [54.5, 57.3] | 38.7 | 15.9 |
| facebook_adv_traine d_resnet152_denoise | 65.3 [64.7, 65.9] | 52.6 [51.2, 54.0] | 38.0 | 39.5 |
| resnet50-randomized_ smoothing_noise_0.50 | 64.2 [63.6, 64.8] | 51.2 [49.8, 52.6] | 39.8 | 61.4 |
| resnet50_with_greyscale_aws | 63.3 [62.7, 63.9] | 50.7 [49.3, 52.1] | 28.3 | 18.7 |
| resnet50_imagenet_subsample_1 _of_4_batch64_original_images | 63.2 [62.6, 63.8] | 50.5 [49.1, 51.9] | 27.4 | 21.4 |
| facebook_adv_traine d_resnet152_baseline | 62.3 [61.7, 63.0] | 49.8 [48.4, 51.2] | 35.8 | 37.6 |
| resnet50-smoothing_adversarial _DNN_2steps_eps_512_noise_0.50 | 62.2 [61.6, 62.8] | 49.1 [47.7, 50.5] | 39.1 | 59.2 |
| resnet50_with_zoom_blur_aws | 61.3 [60.6, 61.9] | 49.4 [48.0, 50.8] | 33.3 | 12.4 |
| shufflenet_v2_x0_5 | 60.6 [59.9, 61.2] | 47.2 [45.8, 48.6] | 23.6 | 16.1 |
| resnet50_adv-train-free | 60.5 [59.9, 61.1] | 47.4 [46.0, 48.8] | 29.4 | 57.4 |
| resnet50-smoothing_adversaria l_PGD_1step_eps_512_noise_0.25 | 60.5 [59.9, 61.1] | 47.0 [45.6, 48.4] | 37.2 | 58.8 |
| resnet50_trained_on_SIN | 60.2 [59.6, 60.8] | 48.6 [47.2, 50.0] | 39.4 | 19.0 |
| squeezenet1_1 | 58.2 [57.6, 58.8] | 45.4 [44.0, 46.8] | 20.2 | 16.1 |
| squeezenet1_0 | 58.1 [57.5, 58.7] | 44.9 [43.5, 46.3] | 20.2 | 18.1 |
| alexnet_lpf2 | 57.2 [56.6, 57.9] | 44.0 [42.6, 45.4] | 22.5 | 29.3 |
| alexnet_lpf3 | 56.9 [56.3, 57.5] | 43.7 [42.3, 45.1] | 22.8 | 30.6 |
| alexnet_lpf5 | 56.6 [56.0, 57.2] | 43.4 [42.0, 44.7] | 22.8 | 32.0 |
| alexnet | 56.5 [55.9, 57.1] | 43.4 [42.0, 44.8] | 21.6 | 24.1 |
| resnet50-smoothing_adversaria l_PGD_1step_eps_512_noise_0.50 | 54.7 [54.0, 55.3] | 41.7 [40.3, 43.1] | 35.7 | 53.8 |

| ImageNet Aggregated Top-1 Accuracies | | | | |
|---|---|---|---|---|
| Model | Validation Accuracy | Imagenetv2 Accuracy | Avg. Corr. Accuracy | Avg. PGD Accuracy |
| resnet50-randomized_ smoothing_noise_1.00 | 53.1 [52.5, 53.7] | 40.7 [39.4, 42.1] | 34.9 | 52.1 |
| resnet50-smoothing_adversarial _DNN_2steps_eps_512_noise_1.00 | 51.9 [51.2, 52.5] | 39.8 [38.4, 41.1] | 34.4 | 51.6 |
| resnet50_imagenet_subsample_1 _of_8_batch64_original_images | 51.7 [51.1, 52.3] | 39.3 [38.0, 40.7] | 19.8 | 19.7 |
| resnet50-smoothing_adversaria l_PGD_1step_eps_512_noise_1.00 | 44.3 [43.7, 44.9] | 33.2 [31.9, 34.6] | 29.8 | 43.7 |
| resnet50_imagenet_subsample_1 _of_16_batch64_original_images | 36.7 [36.1, 37.3] | 27.0 [25.8, 28.3] | 12.7 | 16.0 |
| resnet50_with_defocus_blur_aws | 31.9 [31.3, 32.5] | 24.6 [23.4, 25.8] | 18.2 | 9.6 |
| resnet50_imagenet_subsample_1 _of_32_batch64_original_images | 21.4 [20.9, 21.9] | 15.6 [14.5, 16.6] | 7.2 | 11.1 |

### A.5.1 FULL LIST OF MODELS EVALUATED IN TESTBED

The following list contains all models we evaluated on ImageNet with references and links to the corresponding source code.

1. FixResNeXt101_32x48d_v2 https://github.com/facebookresearch/FixRes Model Type: Trained with more data.

2. FixResNeXt101_32x48d https://github.com/facebookresearch/FixRes Model Type: Trained with more data.

3. instagram-resnext101_32x48d https://github.com/facebookresearch/WSL-Images Model Type: Trained with more data.

4. instagram-resnext101_32x32d https://github.com/facebookresearch/WSL-Images Model Type: Trained with more data.

5. efficientnet-b7 https://github.com/lukemelas/EfficientNet-PyTorch Model Type: Standard training.

6. instagram-resnext101_32x16d https://github.com/facebookresearch/WSL-Images Model Type: Trained with more data.

7. efficientnet-b6 https://github.com/lukemelas/EfficientNet-PyTorch Model Type: Standard training.

8. efficientnet-b5 https://github.com/lukemelas/EfficientNet-PyTorch Model Type: Standard training.

9. FixPNASNet https://github.com/facebookresearch/FixRes Model Type: Standard training.

10. pnasnet5large https://github.com/Cadene/pretrained-models.pytorch Model Type: Standard training.

11. instagram-resnext101_32x8d https://github.com/facebookresearch/WSL-Images Model Type: Trained with more data.

12. efficientnet-b4 https://github.com/lukemelas/EfficientNet-PyTorch Model Type: Standard training.

13. nasnetalarge https://github.com/Cadene/pretrained-models.pytorch Model Type: Standard training.

14. senet154 https://github.com/Cadene/pretrained-models.pytorch Model Type: Standard training.

15. polynet https://github.com/Cadene/pretrained-models.pytorch Model Type: Standard training.

16. efficientnet-b3 https://github.com/lukemelas/EfficientNet-PyTorch Model Type: Standard training.

17. inceptionresnetv2 https://github.com/Cadene/pretrained-models.pytorch Model Type: Standard training.

18. se_resnext101_32x4d `https://github.com/Cadene/pretrained-models.pytorch` Model Type: Standard training.

19. inceptionv4 `https://github.com/Cadene/pretrained-models.pytorch` Model Type: Standard training.

20. resnet101_cutmix `https://github.com/clovaai/CutMix-PyTorch` Model Type: Other robustness intervention.

21. dpn107 `https://github.com/Cadene/pretrained-models.pytorch` Model Type: Standard training.

22. FixResNet50CutMix_v2 `https://github.com/facebookresearch/FixRes` Model Type: Other robustness intervention.

23. dpn131 `https://github.com/Cadene/pretrained-models.pytorch` Model Type: Standard training.

24. FixResNet50CutMix `https://github.com/facebookresearch/FixRes` Model Type: Other robustness intervention.

25. dpn92 `https://github.com/Cadene/pretrained-models.pytorch` Model Type: Standard training.

26. resnext101_32x8d `https://pytorch.org/docs/stable/torchvision/models.html` Model Type: Standard training.

27. efficientnet-b2 `https://github.com/lukemelas/EfficientNet-PyTorch` Model Type: Standard training.

28. dpn98 `https://github.com/Cadene/pretrained-models.pytorch` Model Type: Standard training.

29. google_resnet101_jft-300M Model not publicly released - internal correspondence. Model Type: Trained with more data.

30. se_resnext50_32x4d `https://github.com/Cadene/pretrained-models.pytorch` Model Type: Standard training.

31. resnext101_64x4d `https://github.com/Cadene/pretrained-models.pytorch` Model Type: Standard training.

32. wide_resnet101_2 `https://pytorch.org/docs/stable/torchvision/models.html` Model Type: Standard training.

33. xception `https://github.com/Cadene/pretrained-models.pytorch` Model Type: Standard training.

34. se_resnet152 `https://github.com/Cadene/pretrained-models.pytorch` Model Type: Standard training.

35. resnet50_cutmix `https://github.com/clovaai/CutMix-PyTorch` Model Type: Other robustness intervention.

36. FixResNet50_v2 `https://github.com/facebookresearch/FixRes` Model Type: Standard training.

37. FixResNet50 `https://github.com/facebookresearch/FixRes` Model Type: Standard training.

38. wide_resnet50_2 `https://pytorch.org/docs/stable/torchvision/models.html` Model Type: Standard training.

39. se_resnet101 `https://github.com/Cadene/pretrained-models.pytorch` Model Type: Standard training.

40. resnet152 `https://github.com/Cadene/pretrained-models.pytorch` Model Type: Standard training.

41. resnet50_feature_cutmix `https://github.com/clovaai/CutMix-PyTorch` Model Type: Other robustness intervention.

42. resnext101_32x4d `https://github.com/Cadene/pretrained-models.pytorch` Model Type: Standard training.

43. resnet101_lpf3 `https://github.com/adobe/antialiased-cnns` Model Type: Other robustness intervention.

44. resnet101_lpf5 `https://github.com/adobe/antialiased-cnns` Model Type: Other robustness intervention.

45. efficientnet-b1 `https://github.com/lukemelas/EfficientNet-PyTorch` Model Type: Standard training.

46. resnet101_lpf2 `https://github.com/adobe/antialiased-cnns` Model Type: Other robustness intervention.

47. se_resnet50 `https://github.com/Cadene/pretrained-models.pytorch` Model Type: Standard training.

48. resnext50_32x4d `https://pytorch.org/docs/stable/torchvision/models.html` Model Type: Standard training.

49. resnet50_mixup `https://github.com/clovaai/CutMix-PyTorch` Model Type: Other robustness intervention.

50. fbresnet152 `https://github.com/Cadene/pretrained-models.pytorch` Model Type: Standard training.

51. resnet101 `https://github.com/Cadene/pretrained-models.pytorch` Model Type: Standard training.

52. FixResNet50_no_adaptation `https://github.com/facebookresearch/FixRes` Model Type: Standard training.

53. inceptionv3 `https://github.com/Cadene/pretrained-models.pytorch` Model Type: Standard training.

54. densenet161 `https://github.com/Cadene/pretrained-models.pytorch` Model Type: Standard training.

55. resnet50_cutout `https://github.com/clovaai/CutMix-PyTorch` Model Type: Other robustness intervention.

56. dpn68b `https://github.com/Cadene/pretrained-models.pytorch` Model Type: Standard training.

57. resnet50_lpf5 `https://github.com/adobe/antialiased-cnns` Model Type: Other robustness intervention.

58. densenet201 `https://github.com/Cadene/pretrained-models.pytorch` Model Type: Standard training.

59. resnet50_lpf3 `https://github.com/adobe/antialiased-cnns` Model Type: Other robustness intervention.

60. resnet50_lpf2 `https://github.com/adobe/antialiased-cnns` Model Type: Other robustness intervention.

61. resnet50_trained_on_SIN_and_IN_then_finetuned_on_IN `https://github.com/rgeirhos/texture-vs-shape` Model Type: Other robustness intervention.

62. cafferesnet101 `https://github.com/Cadene/pretrained-models.pytorch` Model Type: Standard training.

63. resnet152-imagenet11k `https://github.com/tornadomeet/ResNet` Model Type: Trained with more data. For this model, we took the 1000 logits corresponding to imagenet classes and computed the accuracies on those logits (as opposed to taking the max over 11k logits).

64. resnet50_aws_baseline `https://github.com/Cadene/pretrained-models.pytorch` Model Type: Standard training.

65. resnet50 `https://github.com/Cadene/pretrained-models.pytorch` Model Type: Standard training.

66. resnet50_imagenet_100percent_batch64_original_images `https://github.com/Cadene/pretrained-models.pytorch` Model Type: Standard training.

67. dpn68 `https://github.com/Cadene/pretrained-models.pytorch` Model Type: Standard training.

68. efficientnet-b0 `https://github.com/lukemelas/EfficientNet-PyTorch` Model Type: Standard training.

69. resnet50-randomized_smoothing_noise_0.00 `https://github.com/locuslab/smoothing` Model Type: Lp adversarially robust. During evaluation, we called `predict` with alpha=1 and n=100.

70. densenet169 `https://github.com/Cadene/pretrained-models.pytorch` Model Type: Standard training.

71. resnet50_with_brightness_aws `https://github.com/Cadene/pretrained-models.pytorch` Model Type: Other robustness intervention.

72. resnet50_with_spatter_aws `https://github.com/Cadene/pretrained-models.pytorch` Model Type: Other robustness intervention.

73. densenet121_lpf3 `https://github.com/adobe/antialiased-cnns` Model Type: Other robustness intervention.

74. densenet121_lpf5 `https://github.com/adobe/antialiased-cnns` Model Type: Other robustness intervention.

75. densenet121_lpf2 `https://github.com/adobe/antialiased-cnns` Model Type: Other robustness intervention.

76. resnet50_with_saturate_aws `https://github.com/Cadene/pretrained-models.pytorch` Model Type: Other robustness intervention.

77. resnet50_trained_on_SIN_and_IN `https://github.com/rgeirhos/texture-vs-shape` Model Type: Other robustness intervention.

78. resnet34_lpf2 `https://github.com/adobe/antialiased-cnns` Model Type: Other robustness intervention.

79. densenet121 `https://github.com/Cadene/pretrained-models.pytorch` Model Type: Standard training.

80. resnet34_lpf3 `https://github.com/adobe/antialiased-cnns` Model Type: Other robustness intervention.

81. vgg19_bn `https://github.com/Cadene/pretrained-models.pytorch` Model Type: Standard training.

82. resnet34_lpf5 `https://github.com/adobe/antialiased-cnns` Model Type: Other robustness intervention.

83. nasnetamobile `https://github.com/Cadene/pretrained-models.pytorch` Model Type: Standard training.

84. vgg16_bn_lpf5 `https://github.com/adobe/antialiased-cnns` Model Type: Other robustness intervention.

85. vgg16_bn_lpf2 `https://github.com/adobe/antialiased-cnns` Model Type: Other robustness intervention.

86. vgg16_bn_lpf3 `https://github.com/adobe/antialiased-cnns` Model Type: Other robustness intervention.

87. resnet50_with_frost_aws `https://github.com/Cadene/pretrained-models.pytorch` Model Type: Other robustness intervention.

88. resnet50_with_jpeg_compression_aws `https://github.com/Cadene/pretrained-models.pytorch` Model Type: Other robustness intervention.

89. bninception `https://github.com/Cadene/pretrained-models.pytorch` Model Type: Standard training.

90. mnasnet1_0 `https://pytorch.org/docs/stable/torchvision/models.html` Model Type: Standard training.

91. vgg16_bn `https://github.com/Cadene/pretrained-models.pytorch` Model Type: Standard training.

92. resnet34 `https://github.com/Cadene/pretrained-models.pytorch` Model Type: Standard training.

93. resnet50_with_gaussian_noise_aws `https://github.com/Cadene/pretrained-models.pytorch` Model Type: Other robustness intervention.

94. resnet50_with_gaussian_noise_contrast_motion_blur_jpeg_compression_aws `https://github.com/Cadene/pretrained-models.pytorch` Model Type: Other robustness intervention.

95. mobilenet_v2_lpf2 `https://github.com/adobe/antialiased-cnns` Model Type: Other robustness intervention.

96. mobilenet_v2_lpf3 `https://github.com/adobe/antialiased-cnns` Model Type: Other robustness intervention.

97. mobilenet_v2_lpf5 `https://github.com/adobe/antialiased-cnns` Model Type: Other robustness intervention.

98. vgg19 `https://github.com/Cadene/pretrained-models.pytorch` Model Type: Standard training.

99. vgg16_lpf5 `https://github.com/adobe/antialiased-cnns` Model Type: Other robustness intervention.

100. vgg16_lpf3 `https://github.com/adobe/antialiased-cnns` Model Type: Other robustness intervention.

101. vgg16_lpf2 `https://github.com/adobe/antialiased-cnns` Model Type: Other robustness intervention.

102. resnet50_with_contrast_aws `https://github.com/Cadene/pretrained-models.pytorch` Model Type: Other robustness intervention.

103. mobilenet_v2 `https://pytorch.org/docs/stable/torchvision/models.html` Model Type: Standard training.

104. resnet50_with_fog_aws `https://github.com/Cadene/pretrained-models.pytorch` Model Type: Other robustness intervention.

105. resnet18_lpf3 `https://github.com/adobe/antialiased-cnns` Model Type: Other robustness intervention.

106. vgg16 `https://github.com/Cadene/pretrained-models.pytorch` Model Type: Standard training.

107. vgg13_bn `https://github.com/Cadene/pretrained-models.pytorch` Model Type: Standard training.

108. resnet18_lpf2 `https://github.com/adobe/antialiased-cnns` Model Type: Other robustness intervention.

109. resnet18_lpf5 `https://github.com/adobe/antialiased-cnns` Model Type: Other robustness intervention.

110. resnet50_imagenet_subsample_1_of_2_batch64_original_images `https://github.com/Cadene/pretrained-models.pytorch` Model Type: Standard training.

111. vgg11_bn `https://github.com/Cadene/pretrained-models.pytorch` Model Type: Standard training.

112. resnet50-randomized_smoothing_noise_0.25 `https://github.com/locuslab/smoothing` Model Type: Lp adversarially robust. During evaluation, we called `predict` with alpha=1 and n=100.

113. vgg13 `https://github.com/Cadene/pretrained-models.pytorch` Model Type: Standard training.

114. googlenet/inceptionv1 `https://pytorch.org/docs/stable/torchvision/models.html` Model Type: Standard training.

115. resnet18 `https://github.com/Cadene/pretrained-models.pytorch` Model Type: Standard training.

116. shufflenet_v2_x1_0 `https://pytorch.org/docs/stable/torchvision/models.html` Model Type: Standard training.

117. vgg11 `https://github.com/Cadene/pretrained-models.pytorch` Model Type: Standard training.

118. resnet50_with_pixelate_aws `https://github.com/Cadene/pretrained-models.pytorch` Model Type: Other robustness intervention.

119. facebook_adv_trained_resnext101_denoiseAll `https://github.com/facebookresearch/ImageNet-Adversarial-Training` Model Type: Lp adversarially robust.

120. resnet50-smoothing_adversarial_DNN_2steps_eps_512_noise_0.25 `https://github.com/Hadisalman/smoothing-adversarial` Model Type: Lp adversarially robust. During evaluation, we called `predict` with alpha=1 and n=100.

121. mnasnet0_5 `https://pytorch.org/docs/stable/torchvision/models.html` Model Type: Standard training.

122. resnet50_with_motion_blur_aws `https://github.com/Cadene/pretrained-models.pytorch` Model Type: Other robustness intervention.

123. facebook_adv_trained_resnet152_denoise `https://github.com/facebookresearch/ImageNet-Adversarial-Training` Model Type: Lp adversarially robust.

124. resnet50-randomized_smoothing_noise_0.50 `https://github.com/locuslab/smoothing` Model Type: Lp adversarially robust. During evaluation, we called `predict` with alpha=1 and n=100.

125. resnet50_with_greyscale_aws `https://github.com/Cadene/pretrained-models.pytorch` Model Type: Other robustness intervention.

126. resnet50_imagenet_subsample_1_of_4_batch64_original_images `https://github.com/Cadene/pretrained-models.pytorch` Model Type: Standard training.

127. facebook_adv_trained_resnet152_baseline `https://github.com/facebookresearch/ImageNet-Adversarial-Training` Model Type: Lp adversarially robust.

128. resnet50-smoothing_adversarial_DNN_2steps_eps_512_noise_0.50 `https://github.com/Hadisalman/smoothing-adversarial` Model Type: Lp adversarially robust. During evaluation, we called `predict` with alpha=1 and n=100.

129. resnet50_with_zoom_blur_aws `https://github.com/Cadene/pretrained-models.pytorch` Model Type: Other robustness intervention.

130. shufflenet_v2_x0_5 `https://pytorch.org/docs/stable/torchvision/models.html` Model Type: Standard training.

131. resnet50_adv-train-free `https://github.com/mahyarnajibi/FreeAdversarialTraining` Model Type: Lp adversarially robust.

132. resnet50-smoothing_adversarial_PGD_1step_eps_512_noise_0.25 `https://github.com/Hadisalman/smoothing-adversarial` Model Type: Lp adversarially robust. During evaluation, we called `predict` with alpha=1 and n=100.

133. resnet50_trained_on_SIN `https://github.com/rgeirhos/texture-vs-shape` Model Type: Other robustness intervention.

134. squeezenet1_1 `https://github.com/Cadene/pretrained-models.pytorch` Model Type: Standard training.

135. squeezenet1_0 `https://github.com/Cadene/pretrained-models.pytorch` Model Type: Standard training.

136. alexnet_lpf2 `https://github.com/adobe/antialiased-cnns` Model Type: Other robustness intervention.

137. alexnet_lpf3 `https://github.com/adobe/antialiased-cnns` Model Type: Other robustness intervention.

138. alexnet_lpf5 `https://github.com/adobe/antialiased-cnns` Model Type: Other robustness intervention.

139. alexnet `https://github.com/Cadene/pretrained-models.pytorch` Model Type: Standard training.

140. resnet50-smoothing_adversarial_PGD_1step_eps_512_noise_0.50 `https://github.com/Hadisalman/smoothing-adversarial` Model Type: Lp adversarially robust. During evaluation, we called `predict` with alpha=1 and n=100.

141. resnet50-randomized_smoothing_noise_1.00 `https://github.com/locuslab/smoothing` Model Type: Lp adversarially robust. During evaluation, we called `predict` with alpha=1 and n=100.

142. resnet50-smoothing_adversarial_DNN_2steps_eps_512_noise_1.00 `https://github.com/Hadisalman/smoothing-adversarial` Model Type: Lp adversarially robust. During evaluation, we called `predict` with alpha=1 and n=100.

143. resnet50_imagenet_subsample_1_of_8_batch64_original_images `https://github.com/Cadene/pretrained-models.pytorch` Model Type: Standard training.

144. resnet50-smoothing_adversarial_PGD_1step_eps_512_noise_1.00 `https://github.com/Hadisalman/smoothing-adversarial` Model Type: Lp adversarially robust. During evaluation, we called `predict` with alpha=1 and n=100.

145. resnet50_imagenet_subsample_1_of_16_batch64_original_images `https://github.com/Cadene/pretrained-models.pytorch` Model Type: Standard training.

146. resnet50_with_defocus_blur_aws `https://github.com/Cadene/pretrained-models.pytorch` Model Type: Other robustness intervention.

147. resnet50_imagenet_subsample_1_of_32_batch64_original_images `https://github.com/Cadene/pretrained-models.pytorch` Model Type: Standard training.

Note about the FixRes models: the github repo for the fixres code uses a slightly different implementation of the `Resize()` method than the PyTorch default. Our testbed was built with the default resizing method, and thus the top1 numbers we report here are around 0.35% lower than what is claimed in the paper. We plan to fix this soon.

## A.6 Details on our evaluation settings

### A.6.1 Real distribution shift

For Imagenetv2, we evaluate on the following datasets: imagenetv2-matched-frequency, imagenetv2-matched-frequency-format-val, imagenetv2-threshold-0.7, imagenetv2-threshold-0.7-format-val, imagenetv2-top-images, imagenetv2-top-images-format-val. The format-val versions are variants of the original dataset encoded with jpeg settings similar to the original one. Unless otherwise stated, results in our paper referring to imagenetv2 are for imagenetv2-matched-frequency-format-val.

For Imagenet-Vid-Robust, we look at the 1109 anchor frames in the dataset and evaluate the benign accuracy for pm0. For pm10, we look at up to 20 nearest frames marked "similar" to the anchor frame in the dataset and count it as a misclassification if any one of the predictions is wrong.

### A.6.2 PGD

We run the following 4 pgd attacks one each model with these settings:

`pgd.linf.eps0.5` Norm: 0.5/255, Step size: 5.88e-5, Num steps: 100

`pgd.linf.eps2` Norm: 2/255, Step size: 2.35e-4, Num steps: 100

`pgd.l2.eps0.1` Norm: 0.1, Step size: 0.01, Num steps: 100

`pgd.l2.eps0.5` Norm: 0.5, Step size: 0.05, Num steps: 100

Most of the models were attacked with only 10% of the dataset (in a class-balanced manner) due to computational constraints. These models are displayed with larger error bars in the plots.

### A.6.3 Corruptions

We include 38 different corruption types: greyscale, gaussian noise (in memory and on disk), shot noise (in memory and on disk), impulse noise (in memory and on disk), speckle noise (in memory and on disk), gaussian blur (in memory and on disk), defocus blur (in memory and on disk), glass blur (in memory), motion blur (in memory and on disk), zoom blur (in memory and on disk), snow (in memory and on disk), frost (in memory and on disk), fog (in memory and on disk), spatter (in memory and on disk), brightness (in memory and on disk), contrast (in memory and on disk), saturate (in memory and on disk), pixelate (in memory and on disk), jpeg compression (in memory and on disk), elastic transform (in memory and on disk).

For each corruption, we average over the five severities. Unfortunately we were not able to implement glass blur efficiently in memory and so that entry is missing. We make sure to make the distinction between in memory corruptions, for which we provide custom fast gpu implementations, and on disk corruptions, for which we use the publicly available imagenet-c dataset, since it was reported in Ford et al. (2019) that jpeg compression can have a significant impact on model accuracies (indeed, as evidenced by Figure 22).

### A.6.4 Stylized Imagenet

We use the stylized imagenet dataset used by (Geirhos et al., 2019) as another evaluation dataset.

### A.6.5 125 class evaluation

For the 125 class evaluation, we evaluate on the following classes from ILSVRC:

```
n01494475 n01630670 n01644373 n01644900 n01669191 n01677366
n01697457 n01742172 n01796340 n01829413 n01871265 n01924916
```

```
n01944390 n01978287 n01980166 n02007558 n02009229 n02017213
n02033041 n02037110 n02056570 n02071294 n02085936 n02086079
n02093428 n02093991 n02095314 n02095570 n02096294 n02096437
n02097474 n02100236 n02100583 n02102318 n02105056 n02107574
n02112706 n02113023 n02114855 n02128925 n02134418 n02138441
n02165105 n02219486 n02226429 n02264363 n02280649 n02441942
n02483708 n02486261 n02488291 n02492035 n02641379 n02730930
n02777292 n02790996 n02795169 n02808440 n02814533 n02814860
n02837789 n02859443 n02892201 n02895154 n02948072 n02951585
n02977058 n03000247 n03110669 n03201208 n03208938 n03216828
n03240683 n03250847 n03272562 n03297495 n03337140 n03376595
n03379051 n03447721 n03492542 n03527444 n03535780 n03642806
n03670208 n03673027 n03692522 n03710193 n03775071 n03832673
n03838899 n03840681 n03868242 n03873416 n03877845 n03884397
n03908714 n03920288 n03933933 n04004767 n04009552 n04037443
n04041544 n04067472 n04074963 n04099969 n04125021 n04141975
n04149813 n04204238 n04208210 n04229816 n04266014 n04310018
n04330267 n04335435 n04336792 n04355338 n04417672 n04479046
n04505470 n07715103 n07875152 n09256479 n12620546
```

