# OpenReview forum: "When Robustness Doesn’t Promote Robustness: Synthetic vs. Natural Distribution Shifts on ImageNet"
_ICLR.cc/2020/Conference — Reject_

### Official Review · AnonReviewer2 · 2019-10-20
**Official Blind Review #2**

**Rating:** 3

**Review:**

Summary: This paper tries to evaluate whether robustness ‘interventions’ such as robustness to adversarial examples and other synthetic distribution shifts on natural distribution shifts.
Overall this paper is generally well written and focuses on an important direction of understanding when the toy/synthetic settings used for convenience actually transfer to the real world. However, I have some concerns with the current version of this work.

Decision: I vote for rejecting this paper. My main concerns are regarding both the motivation of this work and the quality and soundness of the claims.

— Soundness: My major concern is with the datasets that are used for “natural” distribution shift. ImageNetV2 is essentially from the same distribution as the original Imagenet. It’s unclear what the distribution shift is and whether there is a distribution shift. In fact, the paper that produces Imagenet V2 tries very hard to minimize the distribution shift. Hence it seems unreasonable to use this dataset for a shift. Similarly, the other dataset used is from Imagenet Videos. Here again, it’s unclear what the shift actually is, and hence using these datasets to measure performance to natural distribution shifts seems questionable.

— Motivation: I am a little confused about the motivation of this work as presented. It does not seem unreasonable that robustness to one kind of shift doesn’t transfer to another. It seems like the right questions to ask would be the following: Do the synthetic robustness goals seem like they would occur in natural settings? Does robustness on synthetic datasets and synthetic generations of such perturbations transfer to perturbations (of the same kind) but occurring in natural world.

Finally, it seems like the novelty/contribution of this work is very small. The paper uses existing datasets and existing robustness measures.

**Experience Assessment:**

I have read many papers in this area.

**Review Assessment: Checking Correctness Of Derivations And Theory:**

N/A

**Review Assessment: Checking Correctness Of Experiments:**

I assessed the sensibility of the experiments.

**Review Assessment: Thoroughness In Paper Reading:**

N/A

---

> ### Author Response · Authors · 2019-11-10
> **Response**
>
> We thank the reviewer for their comments and respond below to specific concerns:
>
> 1) We disagree with the reviewer’s statement that ImageNetV2 is not distribution shift. The argument made by the ImageNetV2 paper is that despite the authors’ best efforts, ImageNetV2 is a slightly different distribution than the original ImageNet, and this suffices to cause the accuracy drop. Specifically, the authors of the ImageNetV2 paper conclude that
>
> “in spite of our efforts to match the original dataset creation process, the distribution gap is still our leading hypothesis for the accuracy drops.  This demonstrates that it is surprisingly hard to accurately replicate the distribution of current image classification datasets.”  [1]
>
> Hence ImageNetV2 is an interesting and challenging distribution shift for two reasons: 1) the shift is not easily formally characterized, in contrast to adversarial examples and common corruptions; and 2) this shift is a result of a real-world process, rather than a synthetic modification.
>
>
> 2) We agree with the reviewer that it is not a priori clear that adversarial robustness will be helpful outside a fully adversarial setting. Nevertheless, adversarial robustness has been connected to broader claims about robustness in various papers in the literature. Please refer to the list compiled at the end of our response to reviewer 1 for multiple papers making such claims.
>
> Moreover, we also find the question
>
> “Does robustness on synthetic datasets and synthetic generations of such perturbations transfer to perturbations (of the same kind) but occurring in natural world”
>
> very interesting. Including ImageNet-Vid-Robust in our testbed serves precisely this purpose, as the distribution shift is inspired by adversarial examples: the neighborhood of k frames contains images that are visually highly similar, and the evaluation metric is designed as the worst case performance in this neighborhood. This allows us to study comparable distribution shifts from both the synthetic and the natural world perspective. Unfortunately, we find that the synthetic robustness intervention does not meaningfully improve the robustness on the natural distribution shift. We believe that this lack of transfer from synthetic to natural distribution shift demonstrates an important shortcoming of current robustness interventions.
>
> Moreover, we would like to note that distribution shifts occurring in the real world do not always come with a precise characterization, as for example in ImageNetV2. In the absence of a precisely defined distribution shift, it becomes hard or impossible to design corresponding synthetic measures accurately. Thus, a natural question is whether existing robustness interventions at least partially transfer to unforeseen but non-adversarial changes in the data distribution. This is why evaluating current robustness measure on ImageNetV2 is interesting and important.
>
>
> 3) Finally, we would also like to address the reviewer’s comment
>
> “Finally, it seems like the novelty/contribution of this work is very small. The paper uses existing datasets and existing robustness measures.“
>
> We disagree with this characterization of our work. Meta studies such as ours are a backbone of empirical research. The fact that our paper does not introduce new datasets or methods is deliberate and shared by almost all meta studies in the empirical sciences. Meta studies are intended to lead to reliable empirical information, as is especially needed in machine learning today. Rejecting a meta study on grounds that it lacks novel methodology or datasets goes against standard scientific practices.
>
>
>
> [1] Do ImageNet Classifiers Generalize to ImageNet?
> Benjamin Recht, Rebecca Roelofs, Ludwig Schmidt, Vaishaal Shankar
> https://arxiv.org/abs/1902.10811, ICML 2019

---

### Official Review · AnonReviewer3 · 2019-10-22
**Official Blind Review #3**

**Rating:** 6

**Review:**


============================================ Update after rebuttal =========================================

I thank the authors for their detailed rebuttal. I'm in general satisfied with the authors' response to my concerns, so as promised, I'm happy to increase my score.

========================================================================================================

This paper considers the relationship between various measures of synthetic robustness and two distinct measures of natural robustness in large scale image classification models. The authors argue that the synthetic robustness measures considered in this paper are not predictive of natural robustness when the effect of baseline accuracy is subtracted. I think, if true, this is an important message that researchers in this area need to be aware of. However, I have a number of questions and concerns about the results. I would be happy to increase my score if the authors could address some of these issues:

1) Another important recent benchmark not mentioned in the paper is ImageNet-A (https://github.com/hendrycks/natural-adv-examples). I would encourage the authors to include this benchmark among their natural robustness measures (in addition to ImageNetV2 and ImageNetVidRobust). The advantage of this dataset is that because it samples from the error distribution of a high-performing ImageNet model, to a large extent, it already comes with the baseline subtracted, so it essentially obviates the need for the indirect effective robustness measure introduced here. The raw accuracies on ImageNet-A would be directly interpretable and they would also answer the question “why should we care?” in a more visceral way, because even the Instagram trained state-of-the-art ImageNet models seem to achieve a mere 17% accuracy on this benchmark: https://arxiv.org/abs/1907.07640

2) There seems to be a direct conflict between the main conclusion of this paper (that synthetic robustness measures do not predict natural robustness) and an opposite conclusion reached in an earlier paper (https://arxiv.org/abs/1904.10076) where the authors claim that robustness against synthetic perturbations like translation, hue, and saturation are actually highly predictive of video robustness (not sure if this would generalize to ImageNetV2). As far as I can see, these particular perturbation types are not included among the synthetic perturbations considered in this paper. Can you please clarify this discrepancy?

3) Relatedly, looking at the scatter plots of effective robustness vs. robustness against individual perturbations in the appendix, especially for the video robustness measure, some of the correlations seem to be pretty significant (for example, video robustness vs. jpeg compression robustness, p. 19). So, I am wondering to what extent the main conclusion of this paper might just be driven by the averaging of a large number of non-predictive perturbations and a smaller number of more predictive perturbations.

4) Also, ImageNet-P perturbations are not included in the paper. If the authors want to make their claims more reliable, I would encourage them to consider these among their synthetic perturbations as well. The translation perturbation in ImageNet-P, in particular, would be particularly important to consider for video robustness given the results from the arxiv pre-print mentioned in 2) above.

**Experience Assessment:**

I have published one or two papers in this area.

**Review Assessment: Checking Correctness Of Derivations And Theory:**

N/A

**Review Assessment: Checking Correctness Of Experiments:**

I assessed the sensibility of the experiments.

**Review Assessment: Thoroughness In Paper Reading:**

I read the paper at least twice and used my best judgement in assessing the paper.

---

> ### Author Response · Authors · 2019-11-10
> **Response to the first two points raised by the reviewer**
>
> We would like to thank Reviewer 3 for their detailed comments. Below we respond to the specific concerns:
>
>
> 1) Regarding ImageNet-A: First, we would like to emphasize that our testbed is already substantially larger than any previous work on robustness evaluations. Due to size and quick growth of the robustness literature, it is nearly impossible to include all proposed evaluation settings. Hence we had to prioritize some datasets and models over others, and unfortunately some prior work could not be included in our submission. Nevertheless, we believe that our testbed already allows us to draw interesting conclusions about the state of robustness research in machine learning.
>
> We look forward to continuing to maintain and grow our testbed for the community. In particular, we will work on incorporating ImageNet-A, ImageNet-P, hue, and translation perturbations as suggested by the reviewers. Moreover, we welcome suggestions for additional datasets or models for our testbed. However, since these evaluations take many days to run and involve collaborators in industry for some of the models, the results may not be complete in time for the rebuttal period.
>
> Additionally, evaluating on ImageNet-A will not obviate the need for our measure of effective robustness. Though models currently have low accuracy on ImageNet-A, models with higher original accuracy will naturally have higher accuracy on ImageNet-A as well (as can be seen in [1]). This would represent a confounding factor, as it would now be unclear whether the gains in accuracy come from the model being more accurate or whether the gains come from extra robustness properties the model may exhibit. Thus, a metric that corrects for gains coming from simply having a higher original accuracy is preferable. Since empirically we find that original accuracy is an almost perfect predictor of natural distribution shift accuracy for many models (Figure 1), we can measure effective robustness as deviation away from this fit.
>
>
> 2) On synthetic perturbations from [2]: Of the synthetic perturbations mentioned by the reviewer, we currently do have “saturate” in our testbed (please see the plots on page 21, column 2). In our experiments, we find that saturate has no correlation with effective robustness on ImageNet-Vid-Robust. We believe that the discrepancy here is due to different measures of robustness.
>
> In particular, the correlation plots in Figure 4 of [2] do not account for standard accuracy as a confounder. The authors consider video robustness as accuracy within k frames of an anchor frame given that the anchor frame was correctly classified. While this definition does somewhat account for models with higher standard accuracies, it is natural to expect that models with higher standard accuracy are still more likely to predict the neighboring frames correctly given that the anchor frame was predicted correctly. Thus, standard accuracy will be correlated with video robustness. Moreover, standard accuracy is also correlated with corruption accuracy, and hence corruption accuracy will be correlated with video robustness as well.
>
> However, this correlation does not mean that robustness to saturation *causes* robustness on videos. For instance, our testbed contains a model trained on saturation as data augmentation. While this model is highly robust to saturation (exhibiting only a 1% drop from standard accuracy to accuracy under saturations, compared to a baseline model exhibiting a 12% drop), it is no more video robust than a baseline without the saturation training (the saturation-trained model still experiences an 18% video robustness drop, compared to a baseline model exhibiting a 19% drop). This example further shows the need for our measure of effective robustness as it explicitly corrects for the confounding effect of standard accuracy.
>
>
> [1] Hendrycks et al. Natural Adversarial Examples. https://arxiv.org/abs/1907.07174
> [2] Gu et al. Using Videos to Evaluate Image Model Robustness. https://arxiv.org/abs/1904.10076

---

> ### Author Response · Authors · 2019-11-10
> **Response to points 3 and 4 raised by the reviewer**
>
> 3) On video robustness vs JPEG robustness: There is indeed a correlation between the two robustness measures. However, this point is more nuanced since much of the correlation is driven by the lp-adversarially robust models. As pointed out in Section 5 of our paper, the models operate in a low accuracy regime where they are dominated by interpolating between a random classifier and a high accuracy model. So these models do not offer effective robustness in an interesting regime, and hence the correlation between video robustness and JPEG robustness is also substantially diminished.
>
> More broadly, we disagree with the reviewer’s conjecture that “the main conclusion of this paper might just be driven by the averaging of a large number of non-predictive perturbations and a smaller number of more predictive perturbations”. As can be seen from Figure 1, no robustness intervention currently offers substantial effective robustness on the natural distribution shifts we studied. Hence there can also be no highly predictive perturbations since there is simply no robustness to predict.
>
> Nevertheless, we agree with the reviewer that these points and other scatter plots in the appendix of our paper are worth discussing in more detail and will do so in an updated version of the paper.
>
>
> 4) On ImageNet-P: We agree that ImageNet-P would make our testbed more complete, and we will work to incorporate the dataset. But as pointed out in our response to point 1 of the reviewer (see above), it is worth noting that our testbed is already substantially larger than any previous robustness evaluation. We refer the reader to point 1 above for a longer discussion of this issue.

---

### Official Review · AnonReviewer1 · 2019-10-24
**Official Blind Review #1**

**Rating:** 3

**Review:**

This paper studied an interesting question, whether the gain of robustness from synthetic distribution shifts can be transferred/generalized to the robustness under natural distribution shifts. It was shown that in the context of natural distribution shifts, no current robustness intervention can really outperform standard models without a robustness intervention. The main strength of this paper is its extensive experimental study. However, I still have concerns on this work.

1)  The authors mentioned "an implicit assumption underlying this research direction is that robustness to such synthetic distribution shifts will lead to models that also perform more reliably on natural distribution shifts." However, I am uncertain about this point. Let us take adversarial robustness as an example, I am NOT surprising that the robustness on crafted adversarial examples cannot be generalized to the robustness over natural distribution shifts. Thus, I do not think that adversarial robustness obeys the 'implicit assumption'. In other words, the studied problem should be better connected to adversarial robustness.

2) The significance of 'effective robustness' is not clear. It seems that most of insights were learnt from Figure 1, 3 and 4. Do they rely on the metric 'effective robustness'?

3) In Figure 1, 'Trained with more' -> 'Trained with more data'

############# Post-feedback ################
Thanks for the response. I am still not fully convinced by the significance of the findings in this work.
In the paper, the authors highlighted that "our results show that current robustness gains on synthetic distribution shifts do not transfer to improved robustness on the natural distribution shifts presently available as test sets."  I am not surprising at this point, since the synthetic shift, e.g., introduced from adversarial examples, may only characterize the short-cut shift for misclassification. Thus, robustness learnt from this synthetic distribution shift might not transfer to the natural distribution shift.

I decide to keep my original score.


**Experience Assessment:**

I have published in this field for several years.

**Review Assessment: Checking Correctness Of Derivations And Theory:**

N/A

**Review Assessment: Checking Correctness Of Experiments:**

I assessed the sensibility of the experiments.

**Review Assessment: Thoroughness In Paper Reading:**

N/A

---

> ### Author Response · Authors · 2019-11-10
> **The literature has made several connections between adversarial robustness and broader robustness claims.**
>
> We thank the reviewer for their comments and address each point in turn:
>
> 1) We agree with the reviewer that it is not a priori clear that adversarial robustness will be helpful outside a fully adversarial setting. Nevertheless, adversarial robustness has been connected to broader claims about robustness in various papers in the literature. Below, we compile a list of several making such claims and also cite papers connecting non-adversarial corruption robustness to robustness more broadly.
>
>
> 2) Regarding the importance of the effective robustness measure: Figures 1, 3, and 4 do rely on the effective robustness metric. In particular, the scatter plots show the effective robustness as the deviation of various models away from the linear fit indicated by the red line. The figures are interesting precisely because they clearly visualize the effective robustness. An important phenomenon in these three figures is that (almost) all models lie close to the linear fit. In particular, this means that the models have effective robustness close to zero.
>
> Figures 1, 3, and 4 can show the effective robustness via a scatter plot by using both axes as accuracies on two different test sets. While this is useful for visualizing effective robustness as deviation away from the linear fit, it also makes it impossible to plot effective robustness as a function of other quantities (there is no axis left to visualize other quantities). Hence we formally defined the effective robustness measure so we can visualize how effective robustness relates to other quantities such as robustness to various perturbations or to adversarial attacks.
>
> Our response to point 2 of Reviewer 3 offers another example of how effective robustness clarifies whether certain robustness interventions are helpful for making models truly more robust.
>
>
> 3) We will fix the caption of Figure 1.

---

> > ### Author Response · Authors · 2019-11-10
> > **Papers connecting adversarial robustness to broader robustness issues**
> >
> >
> > On Evaluating Adversarial Robustness
> > Nicholas Carlini, Anish Athalye, Nicolas Papernot, Wieland Brendel, Jonas Rauber, Dimitris Tsipras, Ian Goodfellow, Aleksander Madry, Alexey Kurakin
> > https://arxiv.org/pdf/1902.06705.pdf
> >
> > “While there are many valid reasons to study defenses to adversarial examples, below are the three common reasons why one might be interested in evaluating the robustness of a machine learning model. [...] To test the worst-case robustness of machine learning algorithms. Many real-world environments have inherent randomness that is difficult to predict. By analyzing the robustness of a model from the perspective of an adversary, we can estimate the worst-case robustness in a real-world setting. [...] If a powerful adversary who is intentionally trying to cause a system to misbehave (according to some definition) cannot succeed, then we have strong evidence that the system will not misbehave due to any unforeseen randomness.”
> >
> >
> >
> > Adversarial Examples Are a Natural Consequence of Test Error in Noise
> > Nicolas Ford, Justin Gilmer, Nicholas Carlini, Ekin D. Cubuk
> > https://arxiv.org/pdf/1901.10513.pdf, ICML 2019
> >
> > “Over the last few years, the phenomenon of adversarial examples - maliciously constructed inputs that fool trained machine learning models - has captured the attention of the research community, especially when the adversary is restricted to small modifications of a correctly handled input. Less surprisingly, image classifiers also lack human-level performance on randomly corrupted images, such as images with additive Gaussian noise. In this paper we provide both empirical and theoretical evidence that these are two manifestations of the same underlying phenomenon, establishing close connections between the adversarial robustness and corruption robustness research programs. This suggests that improving adversarial robustness should go hand in hand with improving performance in the presence of more general and realistic image corruptions.”
> >
> >
> >
> > A Simple Unified Framework for Detecting Out-of-Distribution Samples and Adversarial Attacks
> > Kimin Lee, Kibok Lee, Honglak Lee, Jinwoo Shin
> > https://arxiv.org/abs/1807.03888, NeurIPS 2018
> >
> > “While most prior methods have been evaluated for detecting either out-of-distribution or adversarial samples, but not both, the proposed method achieves the state-of-the-art performances for both cases in our experiments.”
> >
> >
> >
> > Adversarial Training Versus Weight Decay
> > Angus Galloway, Thomas Tanay, Graham W. Taylor
> > https://arxiv.org/pdf/1804.03308.pdf
> >
> > “The adversarial examples phenomenon afflicting machine learning models has shed light on the need  for  higher  levels  of  abstraction  to  confer  reliable performance in a variety of environments,  where models  may  be  pushed  to  the  limit intentionally by  adversaries,  or unintentionally with  out-of-distribution  data. Although such settings could be seen as violating the i.i.d. assumption implicit in typical machine learning experi-ments, raising this objection provides little comfort when systems fail unexpectedly.”
> >
> >
> >
> > Certifying Some Distributional Robustness with Principled Adversarial Training
> > Aman Sinha, Hongseok Namkoong, John Duchi
> > https://arxiv.org/abs/1710.10571, ICLR 2018
> >
> > “In many systems, robustness to changes in the data-generating distribution P_0 is desirable, whether they be from covariate shifts, changes in the underlying domain [3], or adversarial attacks [24, 31]. As deep networks become prevalent in modern performance-critical systems (perception for selfdriving cars, automated detection of tumors), model failure is increasingly costly; in these situations, it is irresponsible to deploy models whose robustness and failure modes we do not understand or cannot certify.”
> >
> >
> >
> > Towards Deep Learning Models Resistant to Adversarial Attacks
> > Aleksander Madry, Aleksandar Makelov, Ludwig Schmidt, Dimitris Tsipras, Adrian Vladu
> > https://arxiv.org/abs/1706.06083, ICLR 2018
> >
> > “Computer vision presents a particularly striking challenge: very small changes to the input image can fool state-of-the-art neural networks with high confidence [28, 21]. This holds even when the benign example was classified correctly, and the change is imperceptible to a human. Apart from the security implications, this phenomenon also demonstrates that our current models are not learning the underlying concepts in a robust manner.”

---

> > ### Author Response · Authors · 2019-11-10
> > **Papers connecting corruption robustness to broader robustness issues**
> >
> >
> > Benchmarking Neural Network Robustness to Common Corruptions and Perturbations
> > Dan Hendrycks, Thomas Dietterich
> > https://arxiv.org/abs/1903.12261, ICLR 2019
> >
> > “The human vision system is robust in ways that existing computer vision systems are not (Recht et al., 2018; Azulay & Weiss, 2018). Unlike current deep learning classifiers (Krizhevsky et al., 2012; He et al., 2015; Xie et al., 2016), the human vision system is not fooled by small changes in query images. Humans are also not confused by many forms of corruption such as snow, blur, pixelation, and novel combinations of these. Humans can even deal with abstract changes in structure and style. Achieving these kinds of robustness is an important goal for computer vision and machine learning. It is also essential for creating deep learning systems that can be deployed in safety-critical applications.”
> >
> >
> > Generalisation in humans and deep neural networks
> > Robert Geirhos, Carlos R. Medina Temme, Jonas Rauber, Heiko H. Schütt, Matthias Bethge, Felix A. Wichmann
> > https://arxiv.org/abs/1808.08750, NeurIPS 2018
> >
> > “One of the most remarkable properties of the human visual system is its ability to generalise
> > robustly... Even if one never had a shower of confetti before, one is still able to effortlessly recognise objects at a carnival parade. Naturally, such generic, robust mechanisms are not only desirable for animate visual systems but also for solving virtually any visual task that goes beyond a well-confined setting where one knows the exact test distribution already at training time. Deep learning for autonomous driving may be one prominent example: one would like to achieve robust classification performance in the presence of confetti, despite not having had any confetti exposure during training time. Thus, from a machine learning perspective, general noise robustness can be used as a highly relevant example of lifelong machine learning requiring generalisation that does not rely on the standard assumption of independent, identically distributed (i.i.d.) samples at test time.”

---

### Decision · Program_Chairs · 2019-12-19

**Decision:**

Reject

**Comment:**

The authors show that models trained to satisfy adversarial robustness properties do not possess robustness to naturally occuring distribution shifts. The majority of the reviewers agree that this is not a surprising result especially for the choice of natural distribution shifts chosen by the authors (for instance it would be better if the authors compare to natural distribution shifts that look similar to the adversarial corruptions). Moreover, this is a survey study and no novel algorithms are presented, so the paper cannot be accepted on that merit either.